# ADDRESSING DISTRIBUTION SHIFT IN OFFLINE-TO-ONLINE REINFORCEMENT LEARNING

## ABSTRACT

Recent progress in offline reinforcement learning (RL) has made it possible to train strong RL agents from previously-collected, static datasets. However, depending on the quality of the trained agents and the application being considered, it is often desirable to improve such offline RL agents with further online interaction. As it turns out, fine-tuning offline RL agents is a non-trivial challenge, due to *distribution shift* – the agent encounters out-of-distribution samples during online interaction, which may cause bootstrapping error in Q-learning and instability during fine-tuning. In order to address the issue, we present a simple yet effective framework, which incorporates a balanced replay scheme and an ensemble distillation scheme. First, we propose to keep separate offline and online replay buffers, and carefully balance the number of samples from each buffer during updates. By utilizing samples from a wider distribution, i.e., both online and offline samples, we stabilize the Q-learning. Next, we present an ensemble distillation scheme, where we train an ensemble of independent actor-critic agents, then distill the policies into a single policy. In turn, we improve the policy using the Q-ensemble during fine-tuning, which allows the policy updates to be more robust to error in each individual Q-function. We demonstrate the superiority of our method on MuJoCo datasets from the recently proposed D4RL benchmark suite.

## 1 INTRODUCTION

Offline reinforcement learning (RL), the task of training a sequential decision-making agent with a static offline dataset, holds the promise of a data-driven approach to reinforcement learning, thereby bypassing the laborious and often dangerous process of sample collection (Levine et al., 2020). Accordingly, various offline RL methods have been developed, some of which are often capable of training agents that are more performant than the behavior policy (Fujimoto et al., 2019; Kumar et al., 2019; Wu et al., 2019; Siegel et al., 2020; Agarwal et al., 2020; Kidambi et al., 2020; Yu et al., 2020; Kumar et al., 2020). However, agents trained via offline RL methods may be suboptimal, for (a) the dataset they were trained on may only contain suboptimal data; and (b) environment in which they are deployed may be different from the environment in which dataset was generated. This necessitates an online fine-tuning procedure, where the agent improves by interacting with the environment and gathering additional information.

Fine-tuning an offline RL agent, however, poses certain challenges. For example, Nair et al. (2020) pointed out that offline RL algorithms based on modeling the behavior policy (Fujimoto et al., 2019; Kumar et al., 2019; Wu et al., 2019; Siegel et al., 2020) are not amenable to fine-tuning. This is because such methods require sampling actions from the modeled behavior policy for updating the agent, and fine-tuning such a generative model online for reliable sample generation is a challenging task. On the other hand, a more recent state-of-the-art offline RL algorithm, conservative Q-learning (CQL; Kumar et al., 2020), does not require explicit behavior modeling, and one might expect CQL to be amenable to fine-tuning. However, we make an observation that fine-tuning a CQL agent is a non-trivial task, due to the so-called *distribution shift* problem – the agent encounters out-of-distribution samples, and in turn loses its good initial policy from offline RL training. This can be attributed to the bootstrapping error, i.e., error introduced when Q-function is updated with an inaccurate target value evaluated at unfamiliar states and actions. Such initial training instability is a severe limitation, given the appeal of offline RL lies in safe deployment at test time, and losing such safety guarantees directly conflicts with the goal of offline RL.

**Contribution.** In this paper, we first demonstrate that fine-tuning a CQL agent may lead to unstable training due to distribution shift (see Section 3 for more details). To handle this issue, we introduce a simple yet effective framework, which incorporates a balanced replay scheme and an ensemble distillation scheme. Specifically, we propose to maintain two separate replay buffers for offline and online samples, respectively. Then we modulate the sampling ratio between the two, in order to balance the effects of (a) widening the data distribution the agent sees (offline data), and (b) exploiting the environment feedback (online data). Furthermore, we propose an ensemble distillation scheme: first, we learn an ensemble of independent CQL agents, then distill the multiple policies into a single policy. During fine-tuning, we improve the policy using the mean of Q-functions, so that policy updates are more robust to error in each individual Q-function.

In our experiments, we demonstrate the strength of our method based on MuJoCo (Todorov et al., 2012) datasets from the D4RL (Fu et al., 2020) benchmark suite. Our goal is to achieve both (a) strong initial performance as well as maintaining it during the initial training phase, and (b) better sample-efficiency. For evaluation, we measure the final performance and sample-efficiency of RL agents throughout the fine-tuning procedure. We demonstrate that our method achieves stable training, while significantly outperforming all baseline methods considered, including BCQ (Fujimoto et al., 2019) and AWAC (Nair et al., 2020), both in terms of final performance and sample-efficiency.

## 2 BACKGROUND

**Reinforcement learning.** We consider the standard RL framework, where an agent interacts with the environment so as to maximize the expected total return. More formally, at each timestep $t$, the agent observes a state $s_t$, and performs an action $a_t$ according to its policy $\pi$. The environment rewards the agent with $r_t$, then transitions to the next state $s_{t+1}$. The agent's objective is to maximize the expected return $\mathbb{E}_\pi[\sum_{k=0}^\infty \gamma^k r_k]$, where $\gamma \in [0, 1)$ is the discount factor. In this work, we mainly consider off-policy RL algorithms, a class of algorithms that can, in principle, train an agent with samples generated by any behavior policy. These algorithms are well-suited for fine-tuning a pre-trained RL agent, for they can leverage both offline and online samples. Offline RL algorithms are off-policy RL algorithms that only utilize static datasets for training. Here, we introduce an off-policy RL algorithm and an offline RL algorithm we build on in this work.

**Soft Actor-Critic.** SAC (Haarnoja et al., 2018) is an off-policy actor-critic algorithm that learns a soft Q-function $Q_\theta(s, a)$ parameterized by $\theta$ and a stochastic policy $\pi_\phi$ modeled as a Gaussian with its parameters $\phi$. To update parameters $\theta$ and $\phi$, SAC alternates between a soft policy evaluation and a soft policy improvement. During soft policy evaluation, soft Q-function parameter $\theta$ is updated to minimize the soft Bellman residual:

$$\mathcal{L}_{\texttt{critic}}^{\texttt{SAC}}(\theta) = \mathbb{E}_{\tau_t \sim \mathcal{B}}[\mathcal{L}_Q(\tau_t, \theta)], \tag{1}$$

$$\mathcal{L}_Q^{\texttt{SAC}}(\tau_t, \theta) = \left(Q_\theta(s_t, a_t) - (r_t + \gamma \mathbb{E}_{a_{t+1} \sim \pi_\phi}[Q_{\bar{\theta}}(s_{t+1}, a_{t+1}) - \alpha \log \pi_\phi(a_{t+1}|s_{t+1})])\right)^2, \tag{2}$$

where $\tau_t = (s_t, a_t, r_t, s_{t+1})$ is a transition, $\mathcal{B}$ is the replay buffer, $\bar{\theta}$ is the moving average of soft Q-function parameter $\theta$, and $\alpha$ is the temperature parameter. During soft policy improvement, policy parameter $\phi$ is updated to minimize the following objective:

$$\mathcal{L}_{\texttt{actor}}^{\texttt{SAC}}(\phi) = \mathbb{E}_{s_t \sim \mathcal{B}}[\mathcal{L}_\pi(s_t, \phi)], \text{ where } \mathcal{L}_\pi(s_t, \phi) = \mathbb{E}_{a_t \sim \pi_\phi}[\alpha \log \pi_\phi(a_t|s_t) - Q_\theta(s_t, a_t)]. \tag{3}$$

**Conservative Q-Learning.** CQL (Kumar et al., 2020) is an offline RL algorithm that learns a lower bound of the Q-function $Q_\theta(s, a)$, in order to prevent extrapolation error – value overestimation caused by bootstrapping from out-of-distribution actions. To this end, CQL($\mathcal{H}$), a variant of CQL, imposes a regularization that minimizes the expected Q-value at unseen actions, and maximizes the expected Q-value at seen actions by minimizing the following objective:

$$\mathcal{L}_{\texttt{critic}}^{\texttt{CQL}}(\theta) = \frac{1}{2} \cdot \mathcal{L}_Q^{\texttt{CQL}}(\theta) + \alpha^{\texttt{CQL}} \cdot \mathcal{L}_{\texttt{reg}}^{\texttt{CQL}}(\theta), \tag{4}$$

$$\mathcal{L}_Q^{\texttt{CQL}}(\theta) = \mathbb{E}_{\tau_t \sim \mathcal{D}}\left[\left(Q_\theta(s_t, a_t) - (r_t + \gamma \mathbb{E}_{a_{t+1} \sim \pi_\phi}[Q_{\bar{\theta}}(s_{t+1}, a_{t+1})])\right)^2\right], \tag{5}$$

$$\mathcal{L}_{\texttt{reg}}^{\texttt{CQL}}(\theta) = \mathbb{E}_{s_t \sim \mathcal{D}}\left[\log \sum_a \exp(Q_\theta(s_t, a) - \mathbb{E}_{a_t \sim \widehat{\pi}_\beta}[Q_\theta(s_t, a_t)]\right], \tag{6}$$

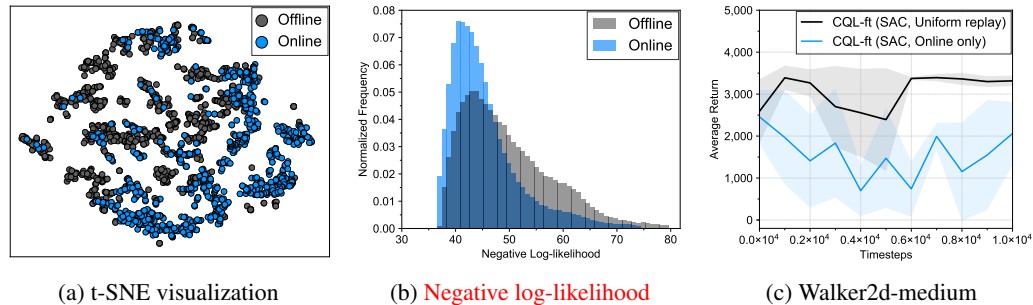

(a) t-SNE visualization      (b) Negative log-likelihood      (c) Walker2d-medium

Figure 1: (a) t-SNE visualization of offline and online samples from the `walker2d-medium` dataset from the D4RL benchmark (Fu et al., 2020), and online samples collected by a CQL agent trained on the same dataset. (b) We visualize the log-likelihood estimates of offline and online samples observed in `walker2d-medium` task, based on a VAE pre-trained on the offline dataset. Offline and online samples follow different distributions. (c) Performance on `walker2d-medium` task during fine-tuning. We first train a CQL agent via (4), and fine-tune it via SAC updates (1), (3), either using both offline and online data drawn uniformly at random (**Uniform replay**), or using online samples only (**Online only**).

where $\alpha^{\texttt{CQL}}$ is the tradeoff parameter, $\mathcal{D}$ the offline dataset, and $\widehat{\pi}_\beta(a_t|s_t) = \frac{\sum_{(s,a)\in\mathcal{D}} \mathbb{1}\{s=s_t, a=a_t\}}{\sum_{s\in\mathcal{D}} \mathbb{1}\{s=s_t\}}$ the empirical behavior policy.

## 3   CHALLENGE: DISTRIBUTION SHIFT

In the context of fine-tuning an offline RL agent, there may exist a distribution shift between offline and online data distribution: an offline RL agent encounters data distributed away from the offline data, as the agent starts interacting with the environment (see Figure 1a for a visualization). More formally, let $p_\pi(s,a) = d^\pi(s)\pi(a|s)$ denote the discounted stationary state-action distribution of the policy, where $d^\pi$ denotes the discounted marginal state distribution, and $\pi$ denotes the policy. Then distribution shift refers to the difference between $p_\beta(s,a)$ and $p_{\pi_\theta}(s,a)$, where $\beta$ refers to the behavior policy, and $\pi_\theta$ refers to the current policy. To demonstrate the existence of distribution shift empirically, we report the negative log-likelihood estimates of offline and online samples in Figure 1b. Specifically, we first split the `walker2d-medium` offline dataset into training and validation datasets, then train a variational autoencoder (VAE; Kingma & Welling, 2014) on the training dataset. Then, we estimate the log-likelihood of online samples and offline samples from the validation dataset, using the importance weight estimator (Burda et al., 2016). We can observe a difference between offline and online data distributions, i.e., they follow different distributions.

This shift in turn affects fine-tuning in a very complicated manner, for it involves an interplay between actor and critic updates with newly collected out-of-distribution samples. To elaborate in more detail, we train an agent offline via state-of-the-art CQL algorithm in (4), then fine-tune it via SAC algorithm in (1) and (3)[1], either using samples drawn uniformly at random (**Uniform replay**) as in Nair et al. (2020), or using online samples exclusively (**Online only**). Figure 1c shows how fine-tuning may suffer from instability in both cases. Essentially, this instability occurs due to the shift between offline and online data distribution. In the case of using both offline and online data, the chance of agent seeing online samples for update becomes too low, especially when the offline dataset contains massive amount of data. This prevents timely updates at unfamiliar states encountered online, and may lead to performance degradation as seen in Figure 1c. On the other hand, when using online samples exclusively, the agent is exposed to unseen samples only, for which Q function does not provide a reliable value estimate. This may lead to bootstrapping error, and hence a dip in performance as seen in Figure 1c. This points to leveraging a mixture of offline and online samples for fine-tuning, so that the agent can balance the trade-off between them.

---

[1] Fine-tuning with CQL updates (4) leads to slow (if at all) improvement in most setups we consider (see Appendix B for more results). We also provide additional experimental results when using different RL algorithms than SAC for fine-tuning a pre-trained offline CQL agent in Appendix C.

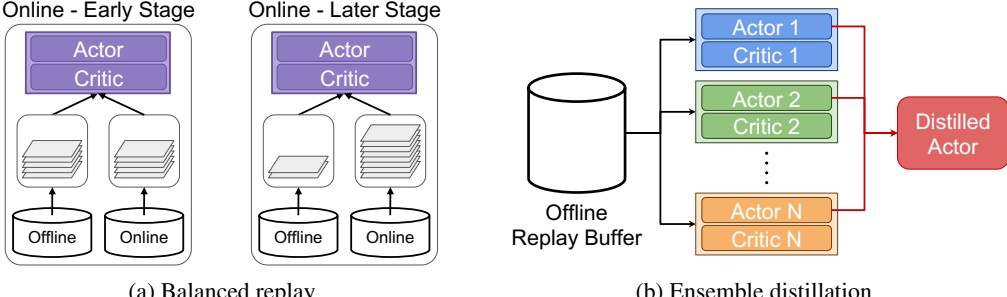

(a) Balanced replay          (b) Ensemble distillation

Figure 2: Illustrations of our framework. (a) For updating the agent during fine-tuning, we draw a balanced number of samples from both offline and online replay buffers initially, but focus on exploiting online samples at a later training stage. (b) We first train an ensemble of $N$ independent offline RL agents with the given offline dataset. Then, we distill the ensemble of policies into a single policy, such that the policy may receive a more accurate learning signal from the Q-ensemble.

## 4   BRED: BALANCED REPLAY WITH ENSEMBLE DISTILLATION

In this section, we present BRED: **B**alanced **R**eplay with **E**nsemble **D**istillation, in order to address the distribution shift between offline and online data distributions. First, we introduce separate offline and online replay buffers in order to select a balanced mix of samples from them for better Q-function updates. This has the dual advantage of updating Q-function with a wide distribution of samples, while making sure that Q-values are updated at novel, unseen states from online interaction. Furthermore, we train multiple actor-critic offline RL agents, and distill the policies into a single policy. Then, during fine-tuning, we improve the distilled policy via Q-ensemble.

### 4.1   BALANCING EXPERIENCES FROM ONLINE AND OFFLINE REPLAY BUFFERS

In order to address the distribution shift between offline and online samples, we introduce separate replay buffers for offline and online samples, respectively. Then, we update the agent using a mix of both offline and online samples. Specifically, at timestep $t$, we draw $B \cdot (1 - \rho_t^{\text{on}})$ samples from the offline replay buffer, and $B \cdot \rho_t^{\text{on}}$ samples from the online replay buffer, where $B$ is the batch size, and the fraction of online samples at timestep $t$, $\rho_t^{\text{on}}$, is defined as follows:

$$\rho_t^{\text{on}} = \rho_0^{\text{on}} + (1 - \rho_0^{\text{on}}) \cdot \frac{\min(t, t_{\text{final}})}{t_{\text{final}}}, \tag{7}$$

where $t_{\text{final}}$ denotes the final timestep of the annealing schedule, and $\rho_0^{\text{on}}$ is the initial fraction of online samples. In other words, we initially let the agent see a mix of both offline samples and online samples, and linearly increase the proportion of online samples (see Figure 2a). This has the effect of (a) better Q-fuction updates with a wide distribution of both offline and online samples, and (b) eventually exploiting the online samples later when there are enough online samples gathered. We empirically show that this simple strategy improves the fine-tuning performance by balancing the experiences from online and offline replay buffers (see Figure 4 for supporting experimental results).

### 4.2   ENSEMBLE OF OFFLINE RL AGENTS FOR ONLINE FINE-TUNING

During fine-tuning, each individual Q-function may be inaccurate due to bootstrapping error from unfamiliar online samples. In order to prevent the policy from learning with such erroneous Q-function, we propose to train multiple actor-critic offline RL agents, then distill the policies into a single policy. In turn, we improve the policy using a more accurate value estimate of the Q-ensemble.

Formally, we consider an ensemble of $N$ CQL agents pre-trained via (4), i.e., $\{Q_{\theta_i}, \pi_{\phi_i}\}_{i=1}^{N}$, where $\theta_i$ and $\phi_i$ denote the parameters of the $i$-th Q-function and $i$-th policy, respectively. We then distill the ensemble of independent policies $\{\pi_{\phi_i}\}_{i=1}^{N}$ into a single distilled policy $\pi_{\phi_{\text{pd}}}$ (see Figure 2b) by

minimizing the following objective before online interaction:

$$\mathcal{L}_{\texttt{distill}}^{\texttt{pd}}(\phi_{\texttt{pd}}) = \mathbb{E}_{s_t \sim \mathcal{D}} \left[ ||\mu_{\phi_{\texttt{pd}}}(s_t) - \widehat{\mu}(s_t)||^2 + ||\sigma_{\phi_{\texttt{pd}}}(s_t) - \widehat{\sigma}(s_t)||^2 \right], \tag{8}$$

$$\widehat{\mu}(s_t) = \frac{1}{N} \sum_{i=1}^{N} \mu_{\phi_i}(s_t), \quad \widehat{\sigma}^2(s_t) = \frac{1}{N} \sum_{i=1}^{N} \left( \sigma_{\phi_i}^2(s_t) + \mu_{\phi_i}^2(s_t) \right) - \widehat{\mu}(s_t)^2, \tag{9}$$

with $\mu_{\texttt{pd}}, \sigma_{\texttt{pd}}^2$ denoting the mean and variance of the distilled policy $\pi_{\phi_{\texttt{pd}}}$, and $\widehat{\mu}, \widehat{\sigma}^2$ the mean and variance of the policy mixture $\frac{1}{N} \sum_{i=1}^{N} \pi_{\phi_i}$. Then, we update the distilled policy by minimizing the following objective instead of (3):

$$\mathcal{L}_{\texttt{actor}}^{\texttt{pd}}(\phi_{\texttt{pd}}) = \mathbb{E}_{s_t \sim \mathcal{B}} \left[ \mathcal{L}_{\pi}^{\texttt{pd}}(s_t, \phi_{\texttt{pd}}) \right], \tag{10}$$

$$\mathcal{L}_{\pi}^{\texttt{pd}}(s_t, \phi_{\texttt{pd}}) = \mathbb{E}_{a_t \sim \pi_{\phi_{\texttt{pd}}}} \left[ \alpha \log \pi_{\phi_{\texttt{pd}}}(a_t|s_t) - \frac{1}{N} \sum_{i=1}^{N} Q_{\theta_i}(s_t, a_t) \right], \tag{11}$$

where $\alpha$ is the temperature parameter. As for Q-function updates, we keep a separate target Q-function $Q_{\bar{\theta}_i}$ for each $Q_{\theta_i}$, then minimize the loss (1) independently, in order to ensure diversity among the Q-functions. Trained this way, the distilled policy receives a more accurate learning signal from the Q-ensemble, although each individual Q-function may be relatively inaccurate. Alternatively, one may also consider an ensemble of independent policies, where actions are samples from $\mathcal{N}(\widehat{\mu}(s_t), \widehat{\sigma}(s_t))$ at timestep $t$, and each policy $\pi_{\phi_i}$ is updated with its corresponding $Q_{\theta_i}$ via (3). However, we show that our ensemble distillation scheme allows for a much more stable improvement over using an ensemble of independent policies in more complex environments (see Figure 5 for supporting experimental results).

## 5   RELATED WORK

**Offline RL.** Offline RL algorithms aim to learn RL agents exclusively with pre-collected datasets. While Agarwal et al. (2020) showed that conventional off-policy RL algorithms work well for large and diverse datasets, it has been observed that Q-function evaluation can be erroneous due to out-of-distribution actions (Fujimoto et al., 2018) otherwise. To address this problem, numerous approaches have been proposed, including policy constraint methods that constrain the policy to stay close to the modeled dataset behavior (Fujimoto et al., 2019; Kumar et al., 2019; Wu et al., 2019; Siegel et al., 2020), and conservative Q-learning methods that train the Q-function to be pessimistic in the unseen regime (Kidambi et al., 2020; Yu et al., 2020; Kumar et al., 2020). Our work builds on a conservative Q-learning method (Kumar et al., 2020), so as to allow fast, unconstrained policy updates once fine-tuning starts.

**Online RL with offline datasets.** Several prior works have explored employing offline datasets for online RL, which offers the advantage of improving both sample-efficiency and exploration. Some assume access to expert demonstrations, then either improve the behavior-cloned policy via on-policy RL (Ijspeert et al., 2003; Kim et al., 2013; Rajeswaran et al., 2018; Zhu et al., 2019), or train a (pre-trained) policy via off-policy RL (Vecerik et al., 2017). These methods are limited, for their success highly depends on the optimality of the dataset. To overcome this, Nair et al. (2020) proposed to train the policy so that it imitates actions with high advantage estimates in offline and online setups alike. However, this method relies on regression, hence the learned policy seldom outperforms the behavior policy that was used to generate the offline dataset. We instead advocate an alternative approach of improving the policy based on the generalization ability of Q-function.

**Replay buffer.** Using replay buffers for managing samples of different categories (e.g., expert and learner, different tasks, offline and online, etc.) has been explored for solving hard exploration problems (Vecerik et al., 2017; Gulcehre et al., 2020), or for continual learning (Rolnick et al., 2019). Our work differs from these, in that we do not assume access to optimal data, nor do we consider multi-task setups. A similar balanced replay scheme as ours has been used in a robot learning setup (Kalashnikov et al., 2018), but their focus is on preventing overfitting. We instead study diverse scenarios with datasets of varying suboptimality, where the agent's initial fine-tuning procedure may suffer due to distribution shift.

**Ensemble methods.** In the context of model-free RL, ensemble methods have been studied for addressing Q-function's overestimation bias (Hasselt, 2010; Hasselt et al., 2016; Anschel et al.,

2017; Fujimoto et al., 2018; Lan et al., 2020), for better exploration (Osband et al., 2016; Chen et al., 2017; Lee et al., 2020), or for reducing bootstrap error propagation (Lee et al., 2020). The closest to our approach is Anschel et al. (2017) that proposed to stabilize Q-learning by using the average of previously learned Q-values as a target Q-value. Our approach instead utilizes an ensemble of independently trained Q-functions, in order to maintain diversity among Q-functions.

## 6 EXPERIMENTS

We designed our experiments to answer the following questions:

- How does BRED compare to other fine-tuning methods (see Figure 3)?
- How crucial is balanced replay for fine-tuning performance (see Figure 4)?
- Can ensemble distillation stabilize the fine-tuning procedure (see Figure 5)?
- How does ensemble size affect performance (see Figure 6)?

### 6.1 SETUPS

**Tasks and implementation details.** We evaluate BRED on MuJoCo tasks (Todorov et al., 2012), i.e., `halfcheetah`, `hopper`, and `walker2d`, from the D4RL benchmark (Fu et al., 2020). The task is to first train an offline RL agent then fine-tuning it . We evaluate the strength of each method based on two criteria: (a) strong initial performance as well as maintaining it, and (b) achieving good sample-efficiency. In order to demonstrate the applicability of our method on various suboptimal datasets, we use four dataset types, i.e., `random`, `medium`, `medium-replay`, and `medium-expert`. Specifically, `random` and `medium` datasets contain samples collected by a random policy and a medium-level policy, respectively. `medium-replay` datasets contain all samples encountered while training a medium-level agent from scratch, and `medium-expert` datasets contain samples collected by both medium-level and expert-level policies. Following the setup in Kumar et al. (2020), we trained offline RL agents for 1000 epochs without early stopping. For our method, we use $N = 5$ for the number of CQL agents, and $t_{\texttt{final}} = 125K$ for the final step of annealing schedule in (7). For the initial fraction of online samples $\rho_0^{\texttt{on}}$, we use the best performing hyperparameter for each dataset by conducting grid search over $\{0.25, 0.5, 0.75, 0.9\}$. More details are provided in Appendix A.

**Baselines**. We consider the following methods as baselines:

- Advantage-Weighted Actor Critic (AWAC; Nair et al., 2020): an actor-critic scheme for fine-tuning, where the policy is trained to imitate actions with high advantage estimates. Comparison of BRED to AWAC shows the benefit of exploiting the generalization ability of Q-function for policy learning.
- BCQ-ft: Batch-Constrained deep Q-learning (BCQ; Fujimoto et al., 2019), is an offline RL method that updates policy by modeling the behavior policy using a conditional VAE (Sohn et al., 2015). We extend BCQ to the online fine-tuning setup by applying the same update rules as offline training.
- CQL-ft: Starting from a CQL agent trained via (4), we fine-tune the agent via SAC updates (1), (3). Justification for excluding the CQL regularization term (6) during fine-tuning can be found in Appendix B.
- SAC: a SAC agent trained from scratch. In particular, we assume the agent does not have access to the offline dataset. We include this baseline to show the benefit of fine-tuning a pre-trained agent in terms of sample-efficiency.

For fairness in comparison, we consider ensembles of baseline methods for comparative evaluation.

### 6.2 COMPARATIVE EVALUATION

Figure 3 shows the performances of BRED and baseline methods during fine-tuning, based on various type of datasets from the D4RL benchmark (Fu et al., 2020). In most tasks, BRED outperforms all baselines in terms of both sample-efficiency and final performance. In particular, our method

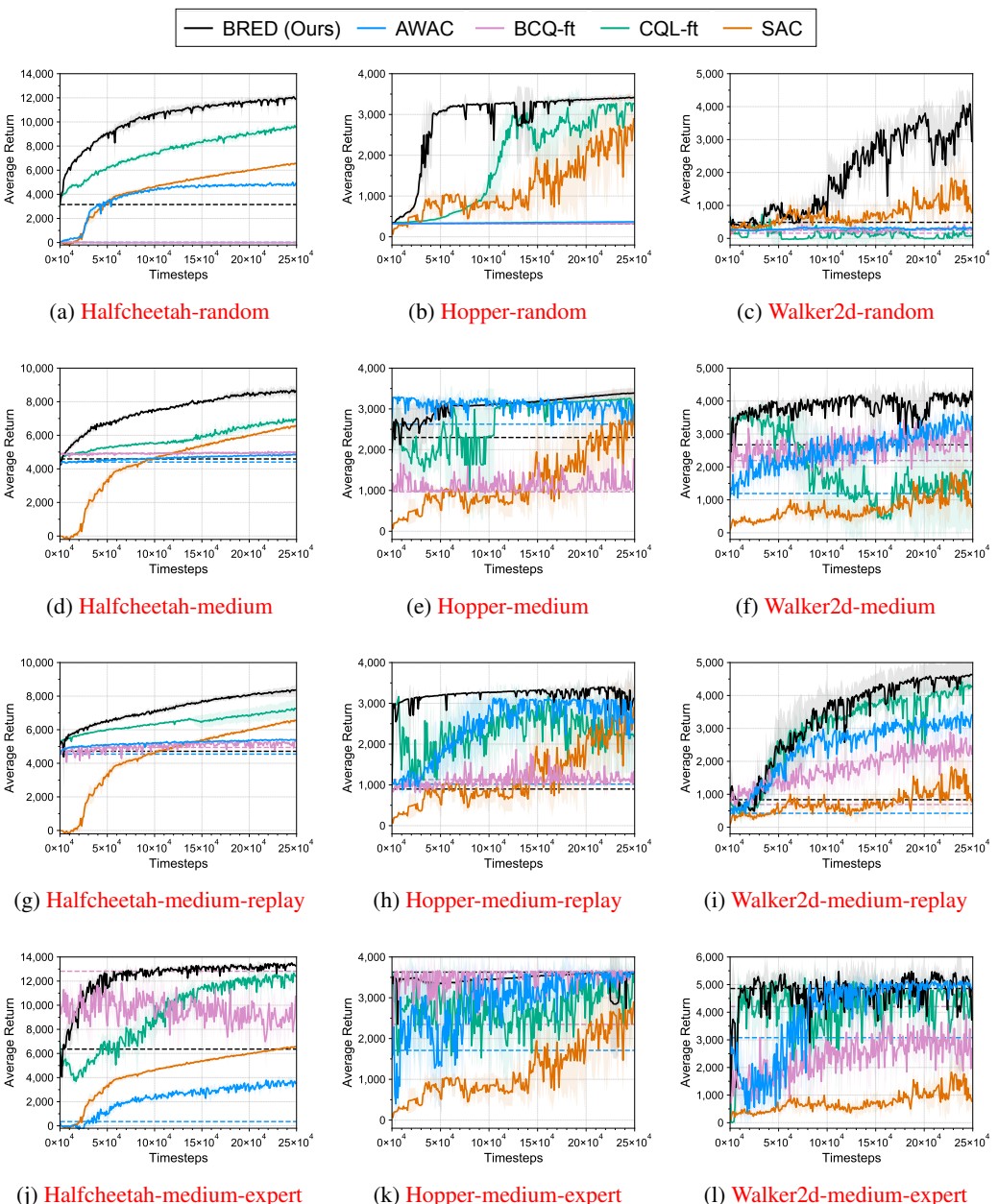

Figure 3: Performance on MuJoCo tasks from the D4RL benchmark (Fu et al., 2020) during online fine-tuning. The solid lines and shaded regions represent mean and 95% confidence interval, respectively, across four runs. Dotted lines indicate the performance of offline RL agents before online fine-tuning.

significantly outperforms CQL-ft, which shows that balanced replay and ensemble distillation play an essential role in stabilizing the fine-tuning process overall. Note that BRED removes the initial performance dip observed in CQL-ft by addressing the harmful effect coming from distribution shift.

We also emphasize that BRED performs consistently well across all tasks, while the performances of AWAC and BCQ-ft are highly dependent on the quality of the dataset used. For example, we observe that AWAC and BCQ-ft show competitive performances in tasks where the datasets are generated by high-quality policies, i.e., `medium-expert` tasks, but perform worse than SAC on `random` tasks. This is because AWAC and BCQ-ft employ the same regularized update rule for offline and online setups alike, and fail to balance the offline and online experiences for replay.

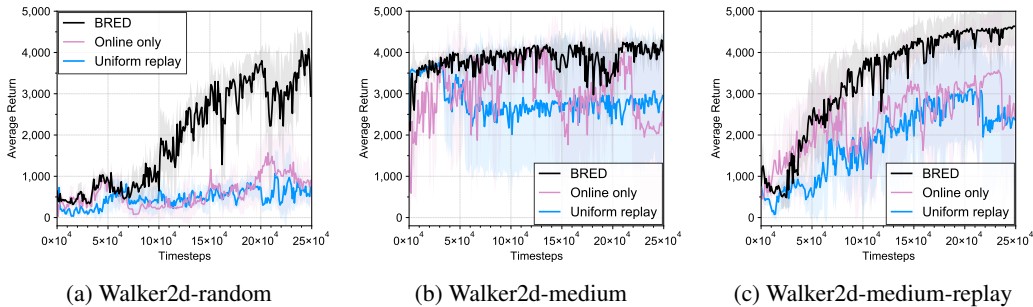

(a) Walker2d-random      (b) Walker2d-medium      (c) Walker2d-medium-replay

Figure 4: Performance on `walker2d` tasks from the D4RL benchmark (Fu et al., 2020) with and without balanced replay. Specifically, we consider two setups: **Uniform replay**, where offline and online samples are sampled uniformly from the same buffer for updates, and **Online only**, where the offline agent is fined-tuned using online samples exclusively.

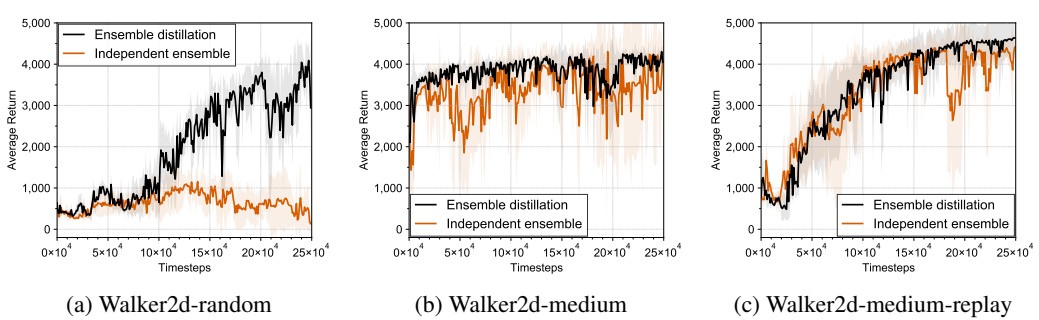

(a) Walker2d-random      (b) Walker2d-medium      (c) Walker2d-medium-replay

Figure 5: We compare BRED (using ensemble distillation) to its variant, an ensemble of independent policies (i.e., no distillation), on `walker2d` tasks. One can observe that ensemble distillation improves both stability and performance during online fine-tuning.

### 6.3 ABLATION AND ANALYSIS

**Effects of balanced replay.** In order to investigate the effectiveness of balanced replay, we compare the performance of BRED with the performance of BRED without balanced replay. In particular, instead of balanced replay scheme, we consider two setups: (i) **Uniform replay**, where offline and online samples are sampled uniformly from the same buffer, and (ii) **Online only**, where the offline agent is fined-tuned using online samples exclusively. For all methods, we applied the proposed ensemble distillation. As seen in Figure 4, BRED is the only method that learns in a stable manner, while the other two methods suffer from slow and unstable improvement. This shows that balanced replay is crucial for addressing distribution shift and stabilizing fine-tuning. Results for all other setups can be found in Appendix D.

**Effects of ensemble distillation.** We also analyze the effect of the proposed ensemble distillation method on fine-tuning performance. To this end, we compare BRED to an ensemble of independent policies trained via (3). As shown in Figure 5, ensemble distillation significantly improves performance in complex tasks such as `walker2d`, where policy updates must be more robust to distribution shift. In particular, for the `walker2d-random` task, the ensemble of independent policies fails to learn a highly performant policy, while BRED achieves a near-expert score at the end. Overall trends are similar for other setups considered (see Appendix E).

**Effects of ensemble size.** We investigate the performance of our method while varying the ensemble size $N \in \{1, 2, 5\}$ on `walker2d-medium-replay` task. We remark that for the single-model case, i.e., $N = 1$, we did not use policy distillation. Figure 6 shows that that the fine-tuning performance of BRED improves as $N$ increases, which shows that larger ensemble size leads to more stable policy updates. More experimental results for all tasks can be found in Appendix F, where the trends are similar.

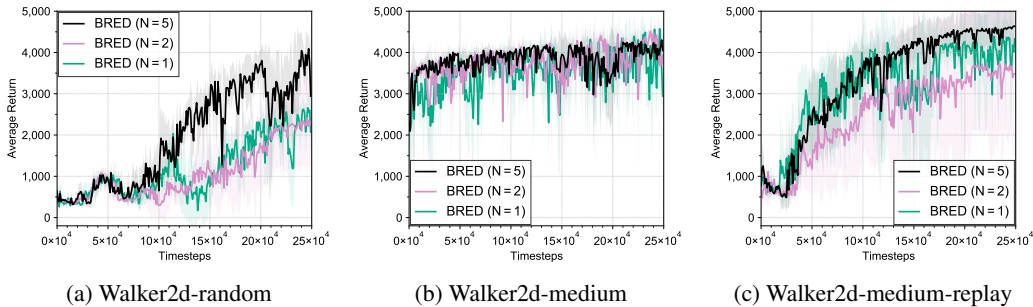

(a) Walker2d-random      (b) Walker2d-medium      (c) Walker2d-medium-replay

Figure 6: Performance on `walker2d` tasks from the D4RL benchmark (Fu et al., 2020) with varying ensemble size $N \in \{1, 2, 5\}$. We observe that performance of BRED improves as $N$ increases.

## 7   CONCLUSION

In this paper, we present BRED, which mitigates the harmful effects of distribution shift between offline and online data distributions, thereby facilitating stable fine-tuning. BRED incorporates two components, namely, a balanced experience replay scheme that mixes offline and online samples for training, and an ensemble distillation scheme for stabilizing policy learning. Our experiments show that BRED performs well across many different setups, and in particular, outperforms prior works that tackle the problem of online reinforcement learning with offline datasets. We believe BRED could prove to be useful for other relevant topics such as sim-to-real transfer (Rusu et al., 2017), scalable RL (Kalashnikov et al., 2018), and RL safety (García & Fernández, 2015).

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

# Appendix

## A EXPERIMENTAL DETAILS FOR BRED

**Offline datasets.** For training offline RL agents with offline datasets, we use the publicly available datasets from the D4RL benchmark (`https://github.com/rail-berkeley/d4rl`) without any modification to the datasets. Note that we use datasets available from the commit `c4bd3de`, i.e., renewed version of datasets.

**Training details for offline RL agents.** For training CQL agents, following Kumar et al. (2020), we built CQL on top of the publicly available implementation of SAC (`https://github.com/vitchyr/rlkit`) without any modification to hyperparameters. As for network architecture, we use 2-layer multi-layer perceptrons (MLPs) for value and policy networks (except for `halfcheetah-medium-expert` task where we found that 3-layer MLPs is more effective for training CQL agents). For training BCQ (Fujimoto et al., 2019), we use the publicly available implementation from `https://github.com/rail-berkeley/d4rl_evaluations/tree/master/bcq`. For all experiments, following the setup in Kumar et al. (2020), we trained offline RL agents for 1000 epochs without early stopping.

**Training details for fine-tuning.** To fine-tune a CQL agent with our method, we initialized parameters of policy and value networks using the parameters from the pre-trained offline CQL agent. For all experiments, we report the performance during the 250K timesteps for 4 random seeds. For our method, we used Adam optimizer (Kingma & Ba, 2015) with policy learning rate of $\{3e-4, 3e-5, 5e-6\}$ and value learning rate of $3e-4$. We found that taking $10\times$ more training steps (10000 as opposed to the usual 1000 steps) after collecting the first 1000 online samples slightly improves the fine-tuning performance. After that, we trained for 1000 training steps every time 1000 additional samples were collected. For AWAC (Nair et al., 2020), we use the publicly available implementation from the authors (`https://github.com/vitchyr/rlkit/tree/master/examples/awac`) without any modification to hyperparameters and architectures. For BCQ-ft, we use the same implementation as the original BCQ, with the only difference being that we used additional online samples for training.

**Training details for balanced replay.** For fine-tuning offline RL agents with the proposed balanced replay, we use $\rho_0^{\mathtt{on}} \in \{0.5, 0.75\}$ for initial fraction of online samples, 256 for batch size $B$, and $t_{\mathtt{final}} = 125K$ for the final step of annealing schedule in (7).

**Training details for ensemble distillation.** For the proposed ensemble distillation method, we use $N = 5$ CQL agents. To distill the ensemble of policies into a single distilled policy, we optimize distillation objective (8) for 1000 epochs with early stopping using the pre-defined validation samples. For action selection during evaluation, we use the mean of distilled policy $\pi_{\phi_{\mathtt{pd}}}$.

## B EFFECTS OF CQL REGULARIZATION ON FINE-TUNING

Here, we show that removing the regularization term (6) from the CQL update is crucial to the fine-tuning performance. As shown in Figure 7, fine-tuning with CQL regularization prevents the agent from improving via online interaction, for the Q-function is updated to be pessimistic for unseen actions. This essentially results in failure to explore. For BRED, we do away with regularization, and instead use the modified SAC update (10).

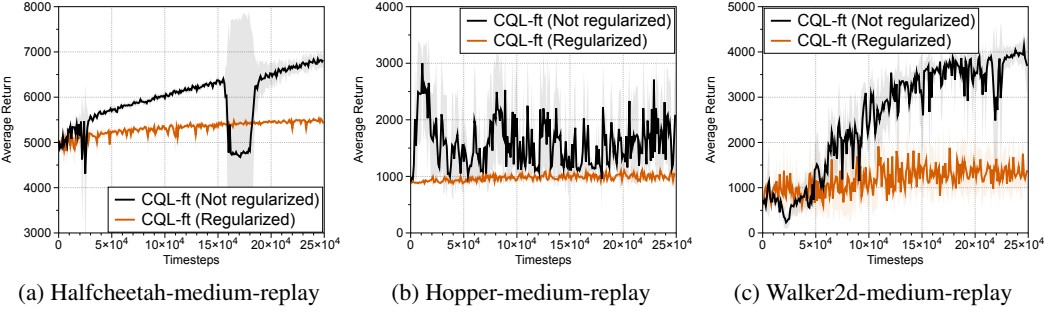

| (a) Halfcheetah-medium-replay | (b) Hopper-medium-replay | (c) Walker2d-medium-replay |

Figure 7: Performance of CQL-ft methods on (a) `halfcheetah-medium-replay`, (b) `hopper-medium-replay`, and (c) `walker2d-medium-replay` tasks from the D4RL benchmark (Fu et al., 2020) during online fine-tuning. The solid lines and shaded regions represent mean and 95% confidence interval, respectively, across four runs.

## C EFFECTS OF ONLINE RL ALGORITHM ON FINE-TUNING

Since SAC optimizes different critic objective (1) which is different from the objective (5) of offline CQL algorithm, this difference in objectives might cause the instability in fine-tuning performance. To investigate how the difference in objective affects the fine-tuning performance, we provide additional experimental results by considering following two online RL algorithms that do not change the critic objective: (i) SAC with deterministic backup (SAC-D) that learns a Q-function by optimizing the same critic objective (5) instead of (1), and (ii) Twin Delayed Deep Deterministic Policy Gradient (TD3; Fujimoto et al., 2018), an off-policy RL algorithm that learns a deterministic policy. As shown in Figure 8b and 8c, agents fine-tuned with SAC-D and TD3 also suffer from instability in the initial phase of fine-tuning, which implies that performance drop is due to the distribution shift problem, not the difference in critic objectives. As shown in Figure 8b and 8c, agents fine-tuned with SAC-D and TD3 also suffer from instability in the initial phase of fine-tuning, which implies that performance drop is due to the distribution shift problem, not the difference in critic objectives.

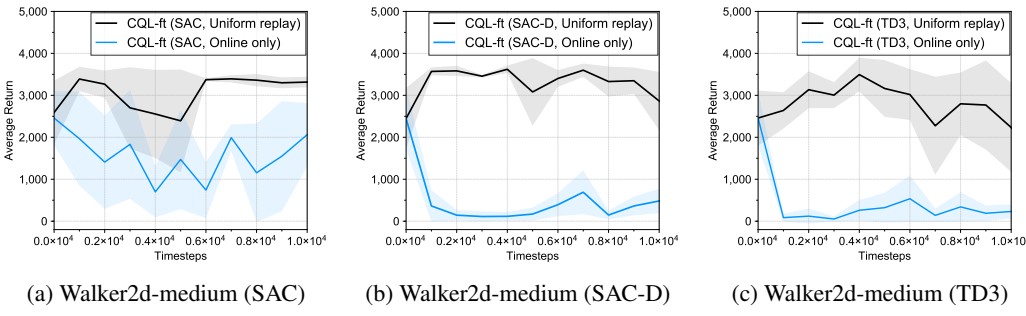

| (a) Walker2d-medium (SAC) | (b) Walker2d-medium (SAC-D) | (c) Walker2d-medium (TD3) |

Figure 8: Fine-tuning performance on `walker2d-medium` task when using (a) SAC, (b) SAC-D, and (c) TD3 for fine-tuning pre-trained offline CQL agents. The solid lines and shaded regions represent mean and 95% confidence interval, respectively, across four runs.

# D EFFECTS OF BALANCED EXPERIENCE REPLAY SCHEME

We provide learning curves for all tasks with and without balanced replay. The proposed balanced replay scheme is indeed crucial for a stable fine-tuning procedure.

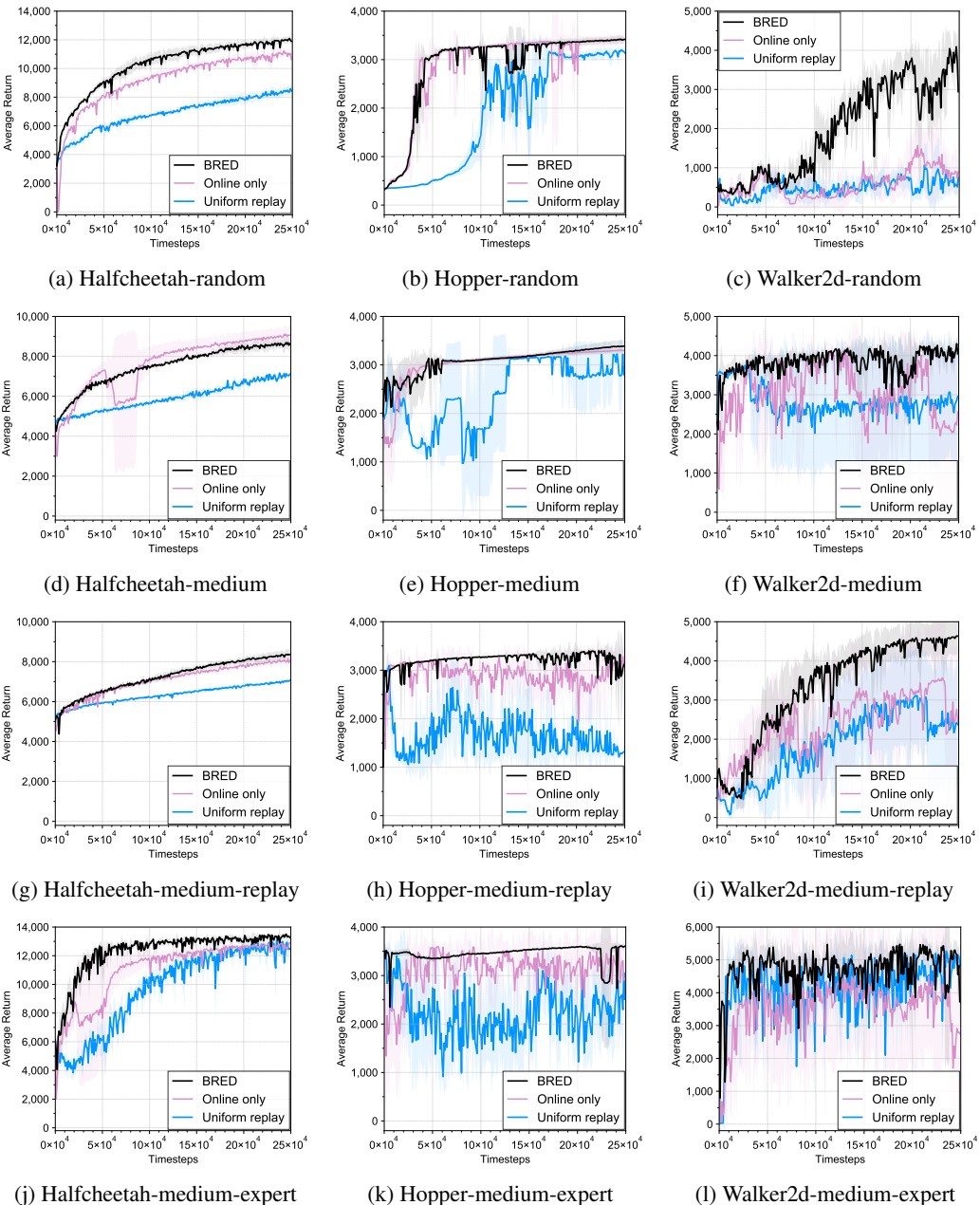

Figure 9: Performance on MuJoCo tasks from the D4RL benchmark (Fu et al., 2020) during online fine-tuning. The solid lines and shaded regions represent mean and 95% confidence interval, respectively, across four runs.

# E    EFFECTS OF ENSEMBLE DISTILLATION

We provide learning curves for all tasks with and without ensemble distillation scheme. One can see that ensemble distillation is highly effective for more complex tasks such as `walker`. However, we observe that BRED without ensemble distillation method, i.e., Independent ensemble, sometimes performs better in simple tasks such as `halfcheetah`, as exploration with diverse policies is relatively more important for these tasks.

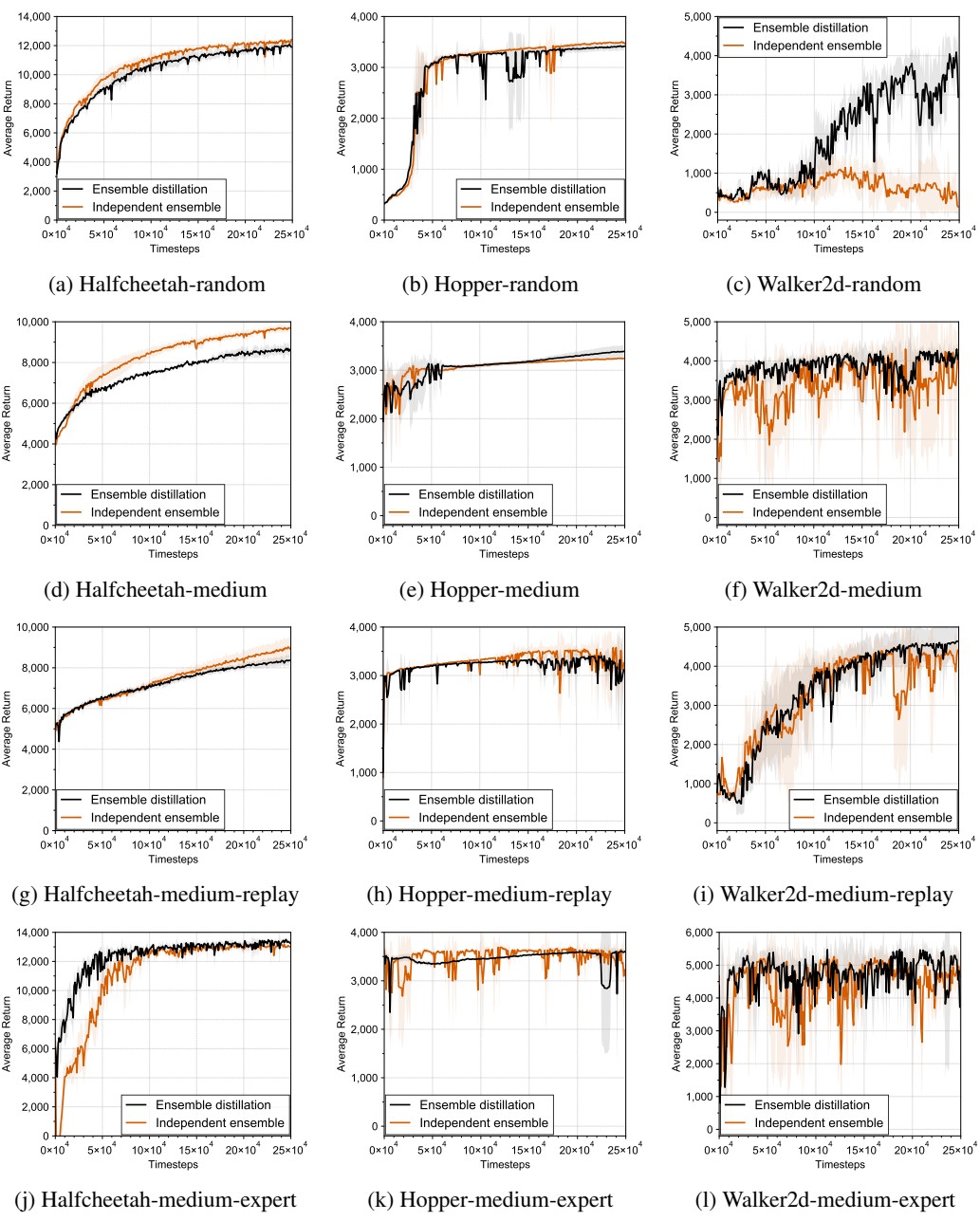

Figure 10: Performance on MuJoCo tasks from the D4RL benchmark (Fu et al., 2020) during on-line fine-tuning. The solid lines and shaded regions represent mean and 95% confidence interval, respectively, across four runs.

# F    EFFECTS OF ENSEMBLE SIZE

We provide learning curves for all tasks with varying ensemble size $N \in \{1, 2, 5\}$. One can see that performance of BRED improves as $N$ grows.

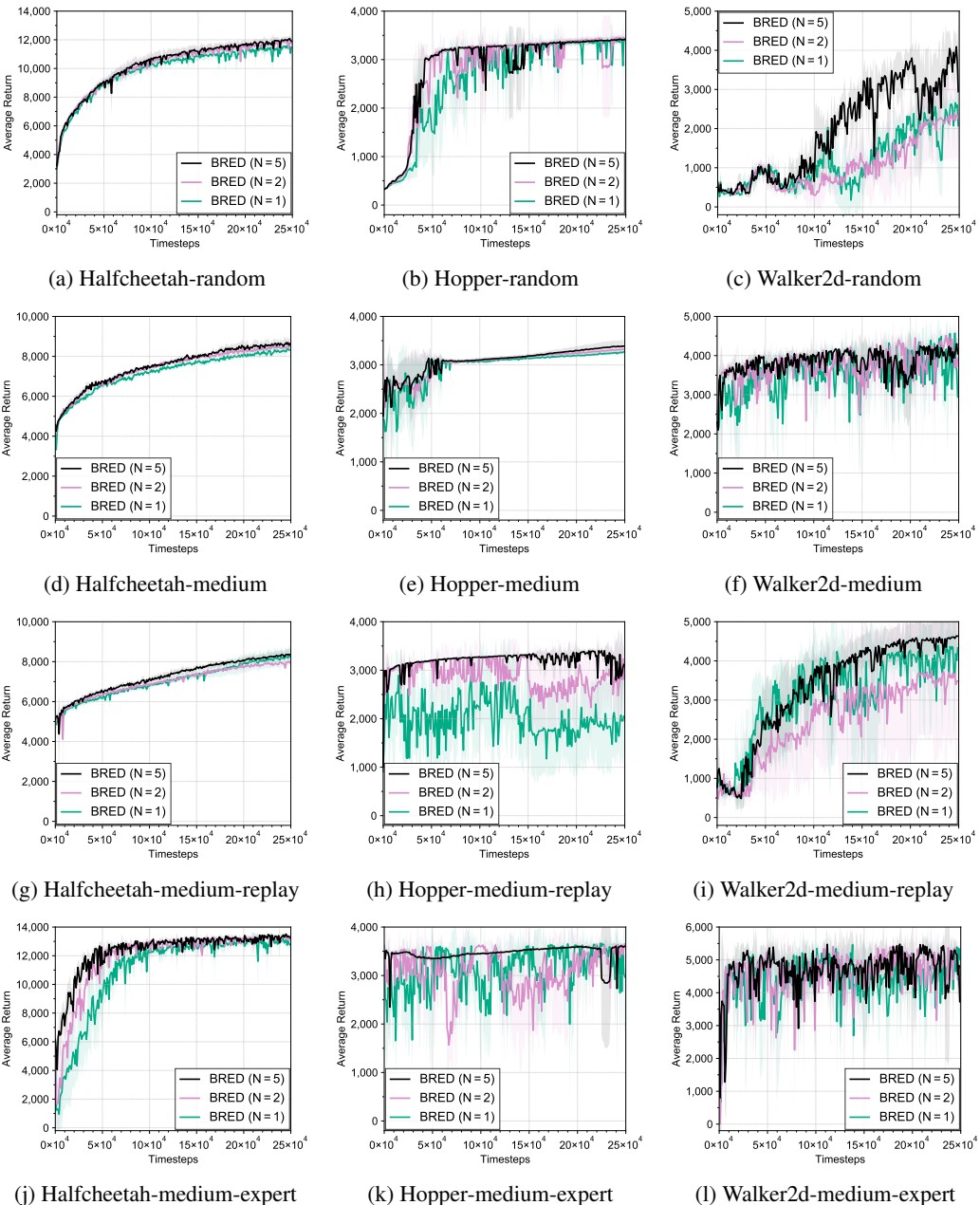

Figure 11: Performance on MuJoCo tasks from the D4RL benchmark (Fu et al., 2020) during on-line fine-tuning. The solid lines and shaded regions represent mean and 95% confidence interval, respectively, across four runs.

# G EFFECTS OF INITIAL FRACTION IN BALANCED REPLAY

In this section, we provide sensitivity analysis of the balanced replay scheme to the initial fraction $\rho_0^{\text{on}}$ in (7). As shown in Figure 12, fine-tuning performance of BRED stays relatively robust to different $\rho_0^{\text{on}}$ in `random`, `medium` tasks, is sometimes sensitive for `medium-replay` and `medium-expert` tasks, where the offline datasets had been generated by a mixture of heterogeneous policies.

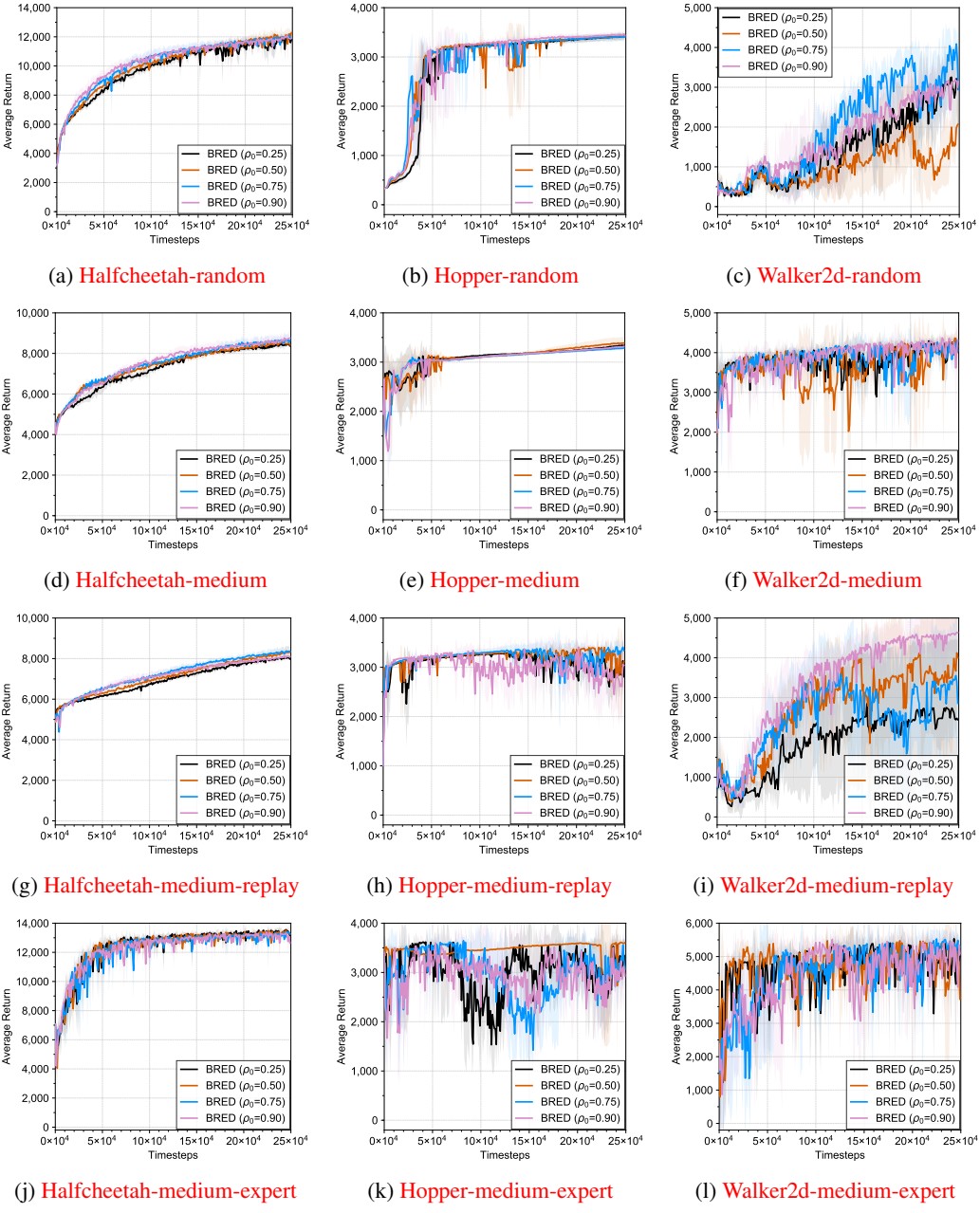

Figure 12: Performance on MuJoCo tasks from the D4RL benchmark (Fu et al., 2020) during online fine-tuning. The solid lines and shaded regions represent mean and 95% confidence interval, respectively, across four runs.

# H  EFFECTS OF FINAL TIMESTEP FOR ANNEALING IN BALANCED REPLAY

In this section, we provide sensitivity analysis of the balanced replay scheme to the final timestep $t_{\texttt{final}}$ in (7). As shown in Figure 13, fine-tuning performance of BRED stays relatively robust to different $t_{\texttt{final}}$ in `random`, `medium` tasks, is sometimes sensitive for `medium-replay` and `medium-expert` tasks, where the offline datasets had been generated by a mixture of heterogeneous policies.

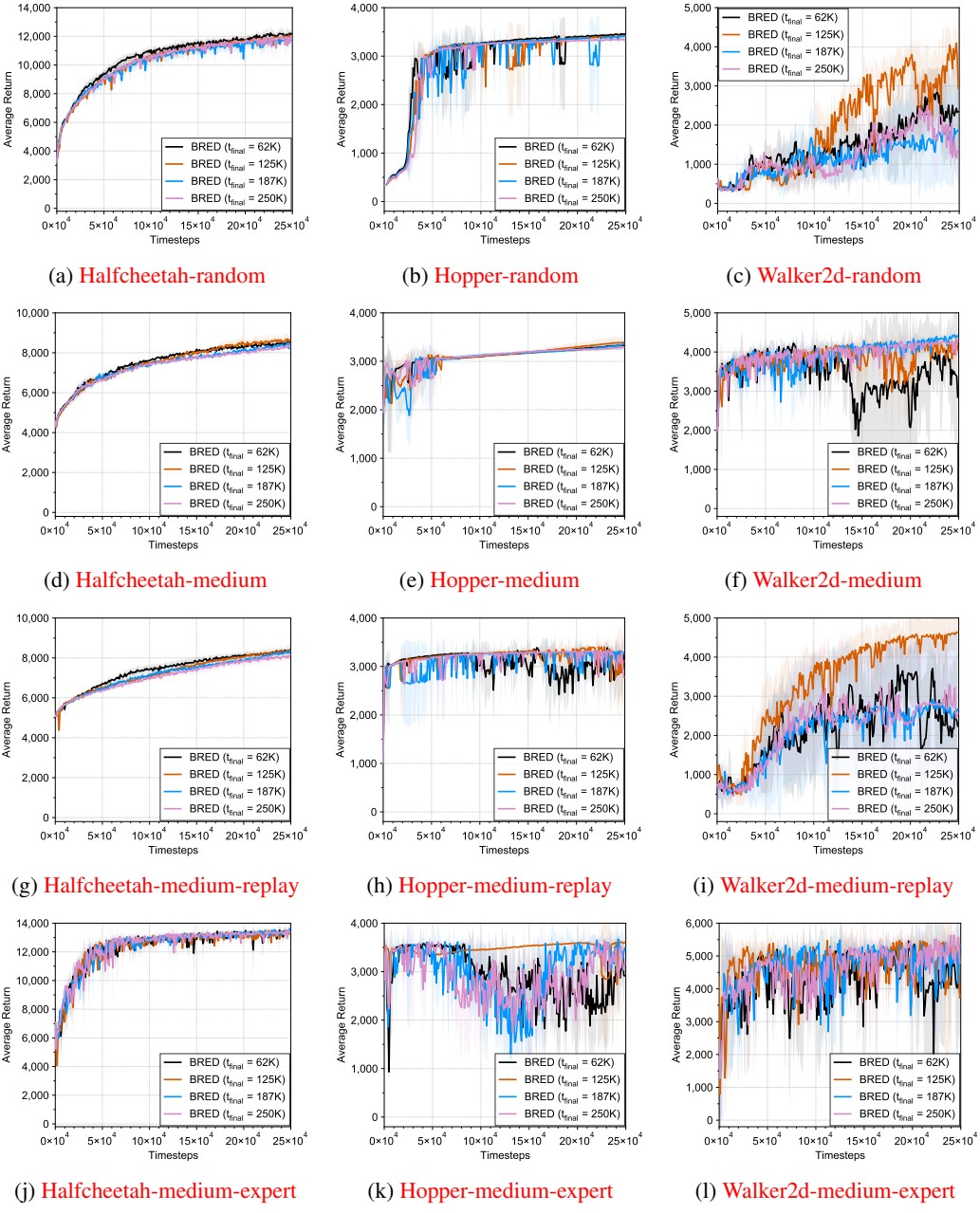

Figure 13: Performance on MuJoCo tasks from the D4RL benchmark (Fu et al., 2020) during online fine-tuning. The solid lines and shaded regions represent mean and 95% confidence interval, respectively, across four runs.

# I EFFECTS OF AN EXPONENTIAL ANNEALING SCHEDULE FOR BALANCED REPLAY

In this section, we provide additional experimental results with different annealing schedule instead of linear annealing schedule in (7). Specifically, we consider exponential annealing schedule:

$$\rho_t^{\text{on}} = \min\left(1,\, \rho_0^{\text{on}}(1+\gamma)^t\right),\text{ where } \gamma = \left(\frac{1}{\rho_0^{\text{on}}}\right)^{t_{\text{final}}^{-1}} - 1$$

Figure 14 shows that exponential annealing schedule sometimes degrades the stability of BRED in tasks such as `walker2d-medium-replay`. We conjecture this is because the fraction of online samples increase abruptly in the middle of online fine-tuning, which leads to unstable Q-learning.

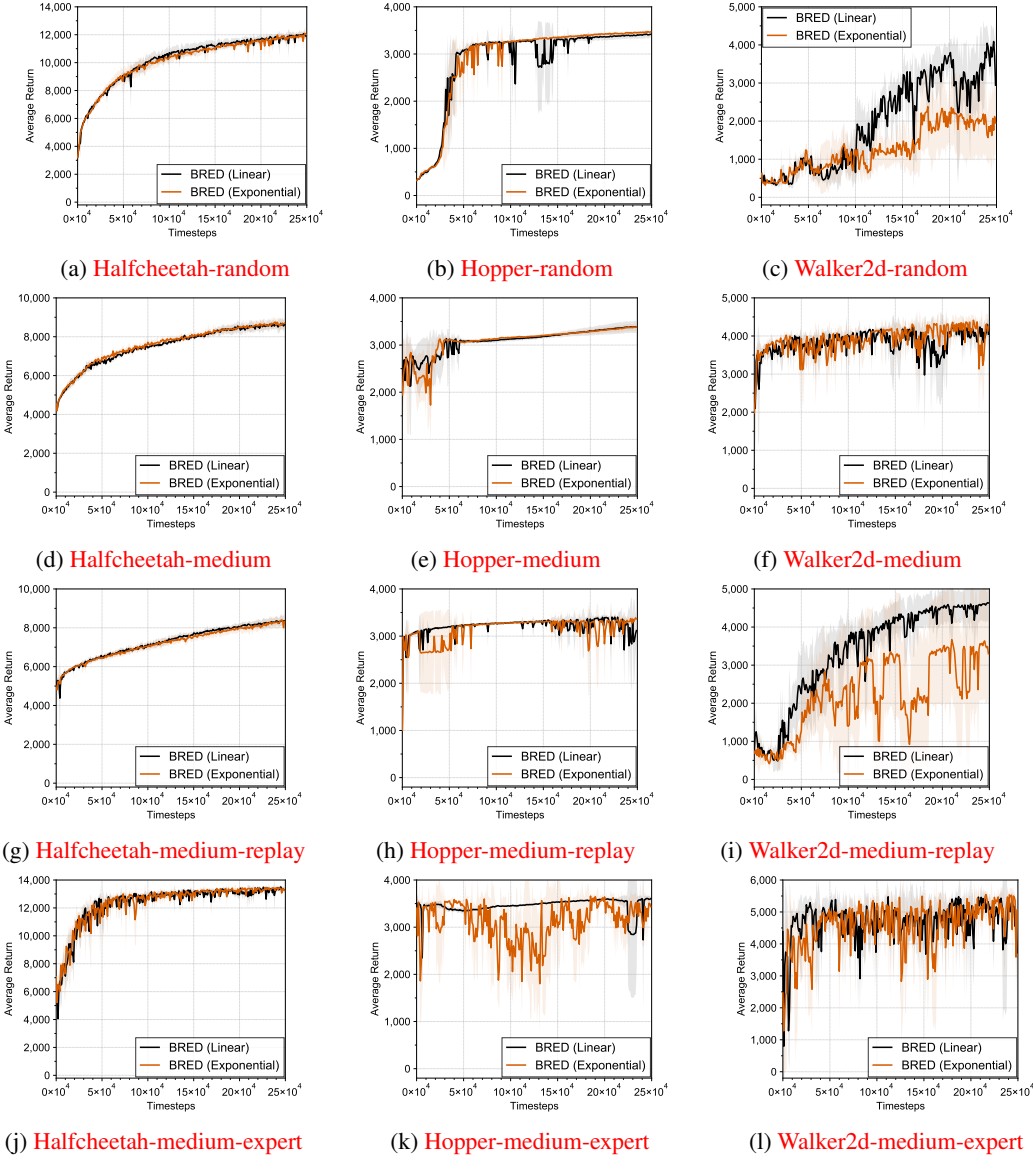

(a) Halfcheetah-random    (b) Hopper-random    (c) Walker2d-random

(d) Halfcheetah-medium    (e) Hopper-medium    (f) Walker2d-medium

(g) Halfcheetah-medium-replay    (h) Hopper-medium-replay    (i) Walker2d-medium-replay

(j) Halfcheetah-medium-expert    (k) Hopper-medium-expert    (l) Walker2d-medium-expert

Figure 14: Performance on MuJoCo tasks from the D4RL benchmark (Fu et al., 2020) during online fine-tuning. The solid lines and shaded regions represent mean and 95% confidence interval, respectively, across four runs.

# J    COMPARATIVE EVALUATION OF UNIFORM REPLAY AGENTS

We further provide ablation studies for BRED, namely, we compare the ensemble distillation agent, with **Uniform replay**. This is to measure the performance of distilled ensemble agent without the balanced replay scheme, i.e. BRED, but without BR. We observe that agents trained as such do not show a particularly strong performance compared to baselines, especially for a difficult control task like `walker2d`. This demonstrates the importance of balanced replay scheme for fine-tuning an offline RL agent.

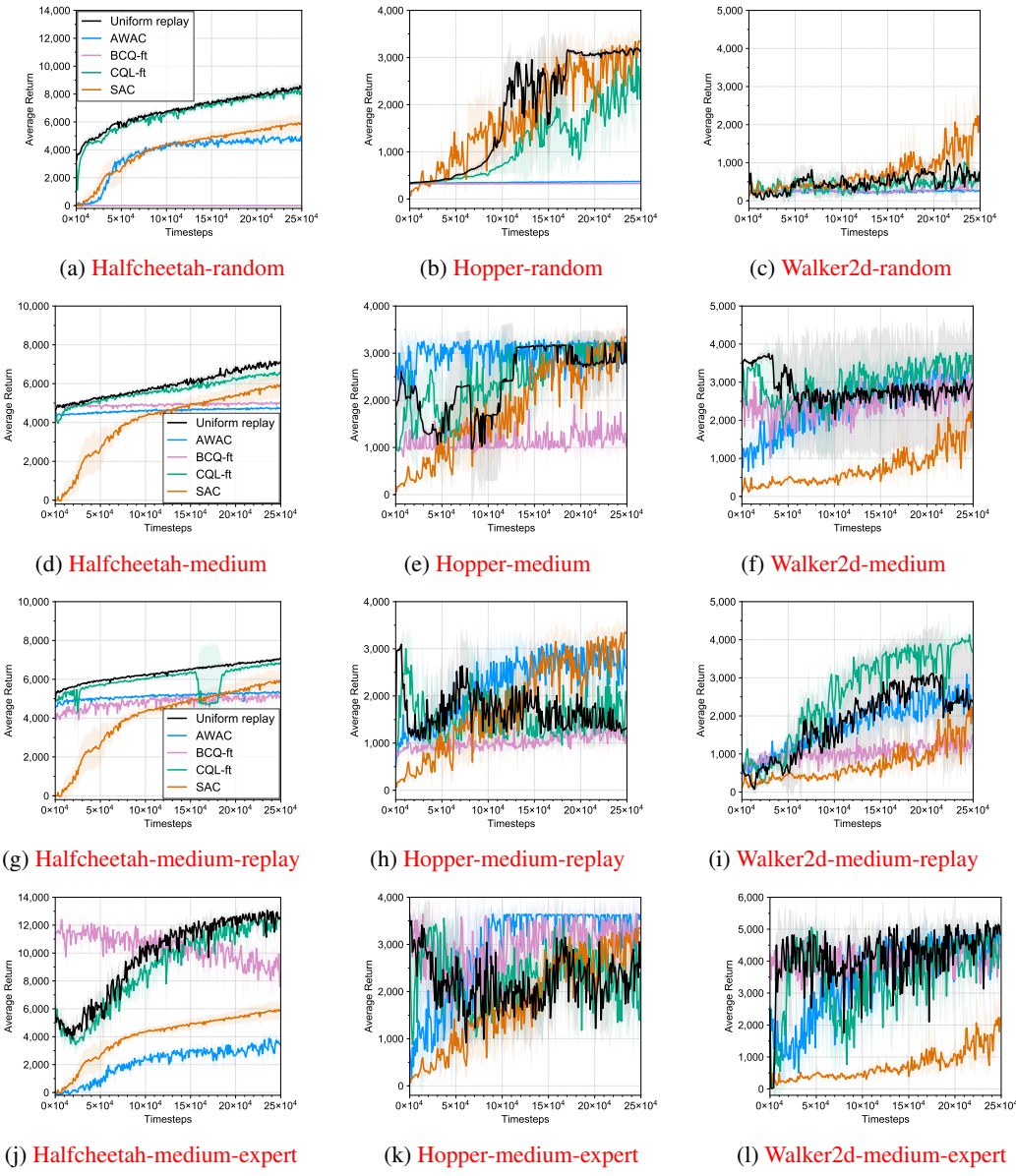

Figure 15: Performance on MuJoCo tasks from the D4RL benchmark (Fu et al., 2020) during on-line fine-tuning. The solid lines and shaded regions represent mean and 95% confidence interval, respectively, across four runs.

# K  EFFECTS OF ENSEMBLE DISTILLATION

Ensemble distillation provides robustness to Q-function approximation error, by reducing policy variance. To show this, for each update during training, given an observation in the sampled mini-batch, we sampled 10 actions from the policy being trained, then measured the variance of Q values across these 10 actions. Then we averaged the variance in Q values across the sampled minibatch. In other words, we calculated the following statistic with mini-batch size 256 and 10 sampled actions: $\mathbb{E}_{s\sim\mathcal{B},a\sim\pi(\cdot|s)}\big[Q_\theta(s,a)\big]$, where we use $\pi_{\phi_{\mathrm{pd}}}$ as $\pi$ for the ensemble distillation agent, and Gaussian policy with mean $\widehat{\mu}$ and variance $\widehat{\sigma}^2$ (9) as $\pi$ for the independent ensemble agent. Figure 16 shows that the variance of Q values for independent ensemble is much higher than for ensemble distillation. This suggests that ensemble distillation indeed provides learning stability.

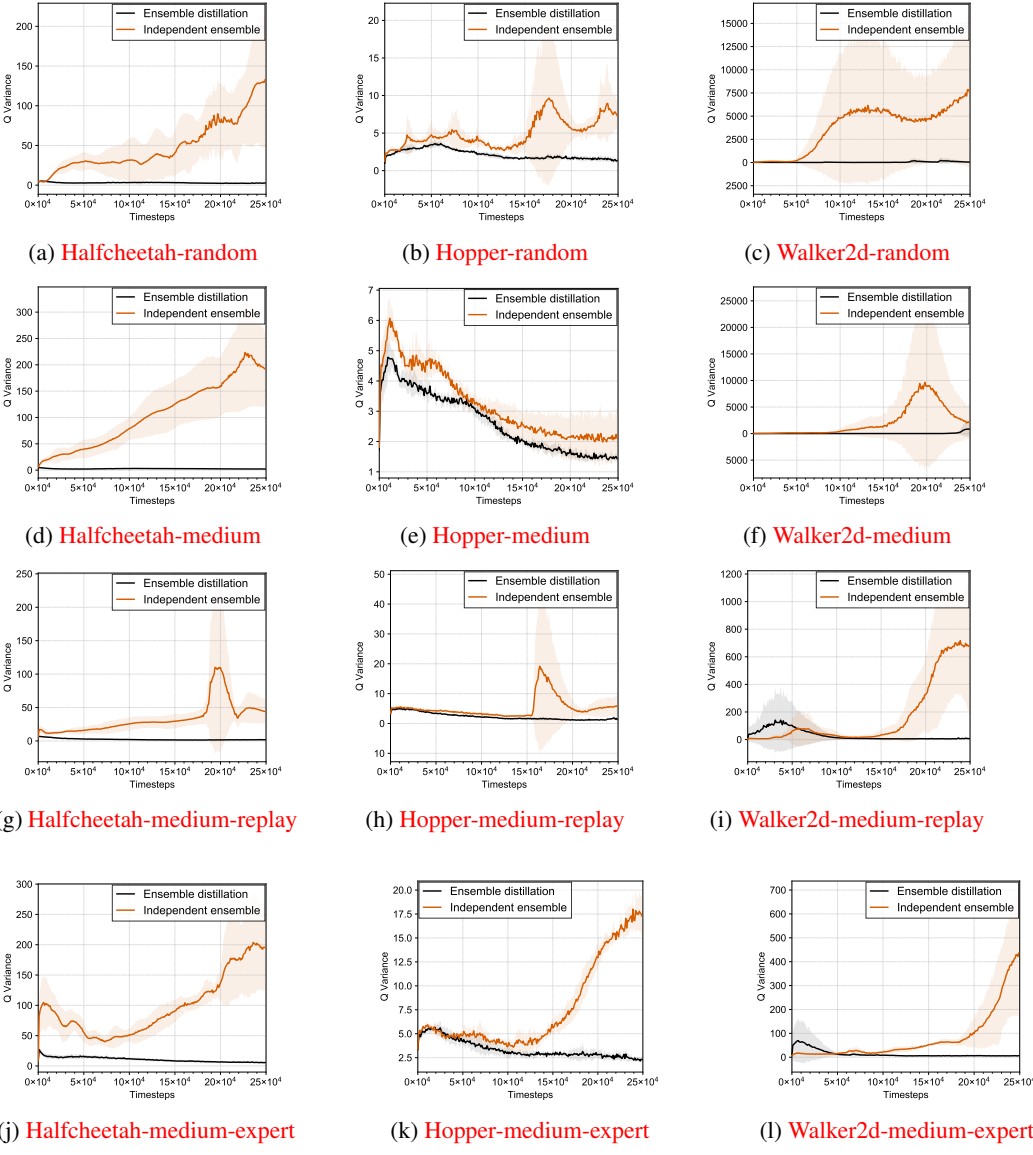

Figure 16: Variance in Q-values for 10 sampled actions averaged across mini-batch during fine-tuning. The solid lines and shaded regions represent mean and 95% confidence interval, respectively, across four runs.

# L    COMPARISON OF Q-FUNCTION ESTIMATES AND TRUE Q FUNCTIONS

In order to demonstrate the harmful effects of distribution shift, and the effectiveness of BRED to deal with such harmful effects, we measured how well the Q-functions of BRED and **Online only** agent represent the ground-true Q fuctions, respectively, in a similar manner to Fujimoto et al. (2018). Here, the **Online only** agent is exactly the same as the BRED agent, except it only uses online samples gathered during fine-tuning. In more detail, at timesteps {10000, 20000, 30000, 40000, 50000}, we randomly sampled 1000 state-action pairs gathered online from the replay buffer, then calculated the average of Q values across these samples. Next, we rolled out 25 episodes starting from each of the state-action pair, then averaged the discounted returns thusly obtained to obtain an estimate of the true Q value at the state-action pair. Finally, we calculated the average of Q value estimates over the 1000 samples, then compared it with the average of Q-function estimates obtained from (1).

As we see in Figure 17, **Online only** agent suffers from high variance and overestimation bias in three out of four tasks considered, namely, `walker2d-random`, `walker2d-medium-replay`, and `walker2d-medium-expert`. On the other hand, BRED shows low-variance Q-function estimates across different runs, and are good estimates of the true Q-function in three out of four tasks, namely, `walker2d-random`, `walker2d-medium-replay`, and `walker2d-medium-expert`. This shows the benefit of balanced replay for stabilizing Q-learning.

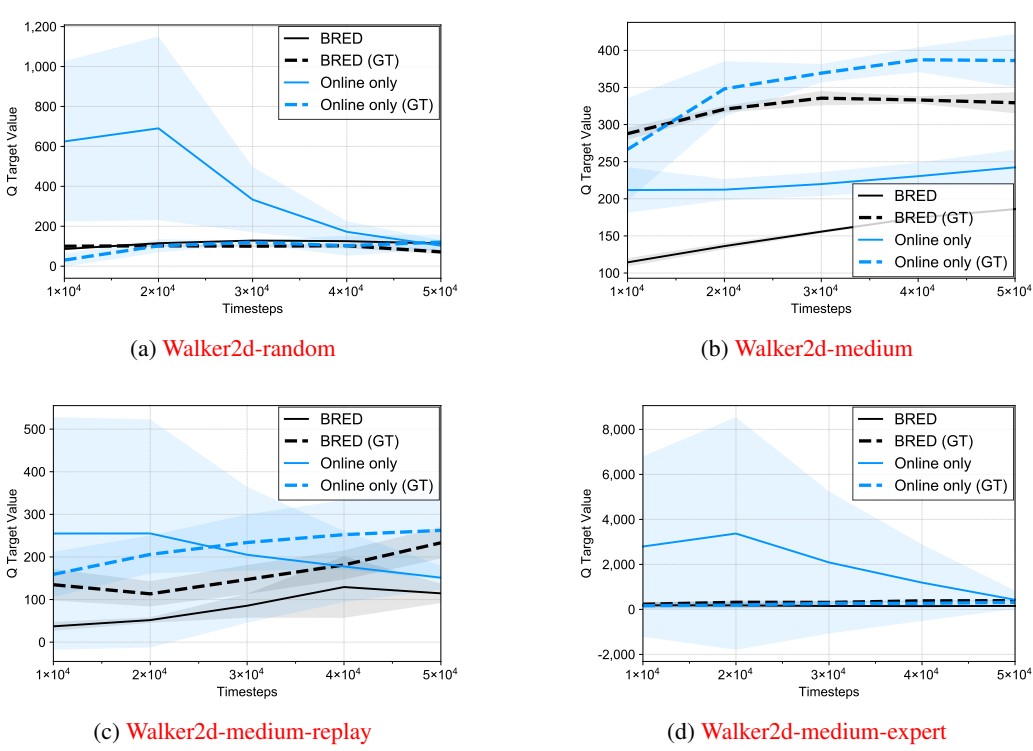

(a) Walker2d-random

(b) Walker2d-medium

(c) Walker2d-medium-replay

(d) Walker2d-medium-expert

Figure 17: Comparison of Q-function estimates and true Q functions for BRED and **Online-only** in Walker2d tasks. We see that BRED shows low variance, and suffers less from overestimation bias.

