# OpenReview forum: "Addressing Distribution Shift in Online Reinforcement Learning with Offline Datasets"
_ICLR.cc/2021/Conference — Reject_

### Official Review · AnonReviewer3 · 2020-10-24
**An interesting work**

**Rating:** 6
**Confidence:** 2

**Review:**

This paper considers the problem of policy learning in Markov Decision Process (MDP) from the combination of online and offline samples. The offline samples are generated by a behavior policy in the same MDP model, i.e., the behavior agent and the learning agent share the same state-action space. The learning procedure goes as follows. One first trains a MDP policy from the offline data; the online samples are then used to fine-tune the learned policy.

The authors propose a simple yet effective approach. First, the authors keep separate offline and online replay buffers, and carefully balance the number of samples from each buffer during updates. Then,  multiple actor-critic offline RL policies are trained, and a single policy is distilled from these policies using ensemble methods. Experiment results show that the proposed method consistently outperforms state-of-art algorithms.

This paper is clearly written and well organized. I am not sure about the novelty of the proposed method, since it seems to follow the line of carefully reweight online and offline samples. However, the experimental results show a significant improvement over existing methods.

Question for the authors:
1. In practice, what is a good heuristic for selecting the initial fraction p0 of online samples? How sensitive is the learned policy w.r.t. the initial fraction p0?

---

> ### Author Response · Authors · 2020-11-19
> **Response to R3**
>
> Dear R3,
>
> We sincerely appreciate your valuable comments and efforts helpful for improving our manuscript. We address each comment in detail, one by one as below.
>
> **(Q1) Novelty of the proposed method.**
>
> (A1) We are aware that various replay schemes and ensemble learning schemes are widely used for RL. However, we believe that applying them in various fine-tuning setups is a novel contribution of our work. For instance, one of the first works in fine-tuning offline RL agents [1] only considers datasets of relatively small sizes (\~100K), in which case the disproportionate sizes of offline dataset and online samples is not a big issue. Accordingly, [1] uses a single replay buffer for fine-tuning. On the other hand, we also consider datasets of disproportionately large sizes (\~1M, \~2M), in which case it is necessary to sample a balanced mix of offline and online samples.
>
> Also, most previous distillation schemes focus on multi-task setups [2,3,4], and most prior works on ensembling in RL focus on ‘train-from-scratch’ scenarios [5,6]. In light of this, we believe our ensemble distillation scheme is a novel contribution to the offline RL & fine-tuning literature.
>
> **(Q2) Initial fraction $\rho_{0}^{\texttt{on}}$.**
>
> (A2) For selecting initial fraction $\rho_{0}^{\texttt{on}}$, we conducted hyperparameter search over $\{0.25, 0.5, 0.75, 0.9\}$ and used the best performing hyperparameter for each dataset. We additionally included the results for these in Figure 12, Appendix G of the revised manuscript. As for sensitivity of the policy to the $\rho_{0}^{\texttt{on}}$, we observe that fine-tuning performance is robust to initial fractions in halfcheetah, and most hopper tasks, but sometimes sensitive in walker2d tasks, which is known to be a difficult control task.
>
> Also, as R4 pointed out, automatic adjustment of sampling ratio is an interesting and promising future direction. For example, one can adjust the sampling ratio based on how distribution of online samples is different from offline data distribution using density esimtation (see (A3) and Figure 1b of the revised manuscript for more details). We would continue to add additional experiments under this scheme, if time permits.
>
> -----------------------------------------------------------------------------------------------------------------------------------------
>
> **REFERENCES**
>
> [1] Nair, Ashvin, et al. "Accelerating online reinforcement learning with offline datasets." arXiv preprint arXiv:2006.09359. 2020.
>
> [2] Rusu, Andrei A., et al. "Policy distillation." International Conference on Learning Representations. 2016.
>
> [3] Schmitt, Simon, et al. "Kickstarting deep reinforcement learning." arXiv preprint arXiv:1803.03835. 2018.
>
> [4] Traoré, René, et al. "DISCORL: Continual reinforcement learning via policy distillation." arXiv preprint arXiv:1907.05855 (2019).
>
> [5] Anschel, Oron, Nir Baram, and Nahum Shimkin. "Averaged-dqn: Variance reduction and stabilization for deep reinforcement learning." International Conference on Machine Learning. 2017.
>
> [6] Osband, Ian, et al. "Deep exploration via bootstrapped DQN." Advances in neural information processing systems. 2016.

---

### Official Review · AnonReviewer1 · 2020-10-26
**Nice paper but some issues need to be addressed**

**Rating:** 4
**Confidence:** 4

**Review:**

##### Summary
The paper proposes a fine-tuning method for an offline RL algorithm, CQL. The method incorporates a balanced replay scheme for both online and offline samples, and an ensemble distillation to stabilize policy learning. The proposed method is evaluated on benchmark environments.

The paper is clearly written and easy to read. However, the problem setting in the paper is not well-motivated and the goal is unclear. The paper makes some claims without supporting them (details come later). Moreover, some important experiments details are missing which makes it hard to assess the empirical results. I think all of these issues needs to be addressed before publication.  Therefore, I recommend to reject the paper.

##### Supporting arguments and clarification questions
First of all, I think it is not motivated why online fine-tuning procedure is necessary (see first paragraph in the introduction). If online simulation/interaction with the real environment is practical, why do we need to train an offline RL agent first and perform online fine-tuning (given that the paper claims that fine-tuning is challenging)? If online simulation is not practical, how could we perform online fine-tuning?

It is also unclear to me what is the goal we want the proposed method to achieve  (e.g., what are the evaluation metrics)? Do we want the fine-tunning method achieve the best sample efficiency compared to purely online algorithm, or achieve the best asymptotic performance compared to other fine-tuning methods? Without a clear goal, it is hard to assess the significance or soundness of the proposed method.

In section 3, what is the reason for fine-tuning a CQL agent by SAC updates? Have you tried fine-tuning by DQN updates or its variant? I think SAC is learning a value function with an extra entropy term and CQL is learning the true value function (with some regularization during the optimization procedure), so these algorithm are essentially learning two different targets. It makes sense that we would see instability during fine-tuning since they are optimizing towards different targets. Therefore, I don’t think the clam “this instability occurs due to the shift between offline and online data distribution” is supported in the experiment. It might be due to the fact that we are optimizing different objectives.

In the experiment, how were the hyper-parameters selected? The hyper-parameters should be selected based on the offline dataset, not the fine-tuning performance. I think the paper should also include a baseline, which is the performance achieved the offline RL agents (i.e., horizontal lines in the learning curve). In Figure 3, it seems like BRED is much more stable compared to other methods in Hopper medium, so I wonder if the reason is that BRED already learns a good policy with offline dataset and it does not change much during the fine-tuning procedures.

In section 5, the paper provides a nice discussion on the related works. However, the paper claims that “this method (Nair et al., 2020) relies on regression, hence the learned policy seldom surpasses the best data-generating policy”. Can you elaborate more on this?

---

> ### Author Response · Authors · 2020-11-19
> **Response to R1 (1/2)**
>
> Dear R1,
>
> We sincerely appreciate your valuable comments for improving our manuscript. We address each comment in detail, one by one as below.
>
> **(Q1) Necessity of online fine-tuning.**
>
> (A1) As R4 mentioned, we believe fine-tuning an offline RL agent is a well-motivated and promising research topic. In particular, fine-tuning a suboptimal offline RL agent is a better strategy than training an agent from scratch when online interaction/simulation is available, for the cost of online interaction usually decreases as the agent becomes more proficient. For one, this is typical of safety-critical tasks such as autonomous driving, where training an agent from scratch would lead to costly failures, whereas an already decent agent would experience relatively infrequent, less costly failures. Also, in robot learning, training is often done in an episodic manner, where a human has to manually reset the agent every time the agent reaches the terminal state, i.e., fails or gets stuck. As pointed out by [1], such manual resets are a serious bottleneck in robot learning. Warm-starting the agent via offline RL then fine-tuning it is one viable strategy to alleviate this issue, and is drawing a lot of research interest accordingly [2,3].
>
> **(Q2) Goal of our work.**
>
> (A2) Our goal is to achieve both (1) strong initial performance as well as maintaining it during the initial training phase, and (2) better sample-efficiency during fine-tuning. We clarified our goal in Section 1 and Section 6.1. Also, to better reflect (1), we added the performance of initial offline RL agents in all relevant performance plots in the revised manuscript.
>
> **(Q3) Fine-tuning a CQL agent with SAC?**
>
> (A3) Thank you for pointing this out. As we have shown in Appendix A of the original manuscript, fine-tuning with the same regularized objective as CQL results in almost no (if at all) improvement of the agent, due to the regularization term inducing pessimistic updates. We used SAC updates for fine-tuning a CQL agent because CQL is built upon SAC in that it learns a stochastic policy and employs an automatic entropy tuning as in SAC. And indeed, with a careful approach, i.e., balanced replay and ensemble distillation, SAC-like updates can lead to stable fine-tuning (Figure 3).
>
> However, to resolve the reviewer’s concern, we have conducted additional experiments using (i) TD3 [4], a more stable variant of DDPG [5], and (ii) SAC [6] with deterministic backup, i.e., with the same objective as CQL except for the regularization term. As shown in Appendix C of the revised manuscript, these methods still suffer from unstable fine-tuning, especially when using online samples exclusively during fine-tuning. This suggests that instability is not due to the difference in optimization objectives, but more due to the distribution shift between offline and online samples.
>
> **(Q4) Hyperparameter selection.**
>
> (A4) We selected the hyperparameters based on grid search, where the selection criteria were sample-efficiency and fine-tuning stability. We agree with the reviewer that it is ideal to select the hyperparameters before fine-tuning, based on the offline dataset and the offline agent trained on it. However, we would like to point out that the task of evaluating an offline RL agent, namely, off-policy evaluation (OPE), is a challenging research question (See [7] for more details). Considering this, deciding on all hyperparameters for fine-tuning a priori is beyond the scope of our work. One rule of thumb we observed to work well, however, is that reduced policy learning rates result in better fine-tuning stability. This observation was also made in a recent work on fine-tuning robotic agents [8]. In addition, we added experimental results concerning different hyperparameter selections for balanced replay scheme in Appendix G and H in the revised manuscript. We observed that BRED is robust to $\rho_{0}^{\tt{on}}$ and $t_{\tt{final}}$ in halfcheetah and most hopper tasks, but sometimes sensitive to them in tasks involving walker2d, which is known to be a difficult control task.

---

> > ### Author Response · Authors · 2020-11-19
> > **Response to R1 (2/2)**
> >
> > **(Q5) BRED already learns a good policy with offline datasets?**
> >
> > (A5) To address this concern, we additionally included the offline performance of BRED (as well as other baselines) in Figure 3. Although we already have strong initial policies for some tasks (hopper-medium-expert, walker2d-medium-expert, for which the offline datasets contain expert-level trajectories), initial policies are usually suboptimal, and fine-tuning is necessary to achieve an expert-level performance.
> >
> > **(Q6) More explanation of AWAC [3]**
> >
> > (A6) Thank you for pointing out a possibly confusing point. For policy learning, AWAC [3] maximizes the following regression objective,
> >
> > $\theta_{k+1} = \arg\max_\theta \mathbb{E}_{s,a \sim \beta}\bigg[ \log \pi_\theta (a|s) \frac1{Z(s)}\exp\bigg( \frac1{\lambda} A^{\pi_k}(s,a) \bigg) \bigg],$
> >
> > i.e., given a state-action pair from the replay buffer, the policy is regressed to the action, where the update rate is weighted by the state-action pair’s advantage estimate. In other words, AWAC essentially performs ‘weighted behavior cloning’, and struggles to extrapolate beyond the policy (or mixture of policies) used to populate the replay buffer. Note that this is reflected in the derivation of AWAC, where they optimize for policy performance, with a constraint that the policy should stay close to the data-generating policy. Indeed, AWAC performs poorly when the dataset quality is suboptimal, e.g., for random datasets.
> >
> > In contrast, BRED performs policy improvement similar to SAC [6], as follows:
> >
> > $\phi_{k+1} = \arg\max_\phi\mathbb{E}_{s \sim \beta, a\sim \pi_\phi(\cdot|s)}\bigg[ \frac1{N} \sum_{i=1}^N {Q_\theta}_i (s,a) - \alpha \log\pi_\phi(a|s) \bigg],$
> >
> > which exploits the generalization/extrapolation abilities of the Q-function, and hence results in a faster fine-tuning procedure. We added more clarification in Section 5 of the revised manuscript.
> >
> > -----------------------------------------------------------------------------------------------------------------------------------------
> >
> > **REFERENCES**
> >
> > [1] Zhu, Henry, et al. "The Ingredients of Real World Robotic Reinforcement Learning." International Conference on Learning Representations. 2019.
> >
> > [2] Kalashnikov, Dmitry, et al. "Qt-opt: Scalable deep reinforcement learning for vision-based robotic manipulation." Conference on Robot Learning. 2018.
> >
> > [3] Nair, Ashvin, et al. "Accelerating online reinforcement learning with offline datasets." arXiv preprint arXiv:2006.09359 (2020).
> >
> > [4] Fujimoto, Scott, Herke Hoof, and David Meger. "Addressing Function Approximation Error in Actor-Critic Methods." International Conference on Machine Learning. 2018.
> >
> > [5] Lillicrap, Timothy P., et al. "Continuous control with deep reinforcement learning." International Conference on Learning Representations. 2016.
> >
> > [6] Haarnoja, Tuomas, et al. "Soft Actor-Critic: Off-Policy Maximum Entropy Deep Reinforcement Learning with a Stochastic Actor." International Conference on Machine Learning. 2018.
> >
> > [7] Anonymous. “Benchmarks for deep off-policy evaluation.” In Submitted to International Conference on Learning Representations, 2021. URL https://openreview.net/forum?id=kWSeGEeHvF8. under review.
> >
> > [8] Julian, Ryan, et al. “Never Stop Learning: The Effectiveness of Fine-Tuning in Robotic Reinforcement Learning.” Conference on Robot Learning. 2020.

---

### Official Review · AnonReviewer4 · 2020-10-26
**The idea is good, but more work is needed.**

**Rating:** 5
**Confidence:** 4

**Review:**

Summary:
This paper proposes to deal with distribution shift problem between online and offline samples when the agent trained by offline data is fine-tuned with online interactions. Two mechanisms are introduced: (1) using two replay buffers for offline and online data respectively, and training the agent with data sampled from these two buffers with a certain ratio (the ratio changes in a way that more online data is used in later epochs); (2) learning an ensemble of independent agents in the offline phase, and distilling them into a mean policy to overcome bootstrapping error. Empirical results demonstrate that the proposed method perform well during fine-tuning when there is distribution shift.


Strengths:
1. The paper is overall well-written and easy to follow.
2. The problem studied by this paper is well-motivated, and the proposed methods are simple yet effective, making intuitive sense.
3. The experiment section shows that the proposed method (BRED) outperforms baselines on 3 mujoco tasks with multiple types of offline data. Further ablation study verifies the effectiveness of the two proposed mechanisms.


Weakness:
1. Although empirical results are good, my main concern is that the novelty of this paper is limited. The main contribution of this paper is combining CQL with two tricks: balanced replay and ensemble distillation. As already discussed in the related work section, these two tricks are commonly used in the literature, even though the concrete setting varies.
2. The specific way of combining the offline buffer and the online buffer (online fraction grows linearly in timestep) could be effective in some cases, but might not always be the best choice. I am curious whether the authors have tried other ways. And especially, can the agent chooses the ratio adaptively based on the degree of distribution shift and the learning performance?
3. The definition of distribution shift in this paper is descriptive and informal. It will be nice to provide deeper analysis or insights w.r.t. the distribution shift problem. For example, is there a way to rigorously characterize the distribution shift? Is there a measurement for how large the gap is? This will be relevant to addressing point 2 above.
4. Although policy ensemble can make the learned Q-function more accurate and stable, it is not very clear to me why it tackles distribution shift. Even the distilled policy has a more accurate estimation of the Q values, the estimation is still w.r.t. the offline data distribution. It still suffers from the bootstrapping error when using out-of-distribution samples.


Minor comments:
1. Figure 1 is a good visualization. However, the figure is not very informative since details are not provided (e.g., what is contained in a sample, in what sense / which dimension the distribution is different).
2. In the experiment, I am wondering how an “ensemble-only” method would work and compare to baselines, i.e., use multiple pertained CQL agents, and fine-tune the distilled policy. The “Effects of ensemble distillation” paragraph shows a comparison between the proposed ensemble distillation and the ensemble of independent policies. However, the result does not reflect how ensemble compares with non-ensemble.
3. The paper focuses on the distribution shift problem in offline RL. However, there could be multiple interpretations of distribution shift, e.g., state distribution changes due to the change of behavior policy, or dynamics distribution changes due to the slight change of the environment, etc. I think this paper is mainly dealing with the state distribution change case. But it will be better if the authors can make it explicit.

---

> ### Author Response · Authors · 2020-11-19
> **Response to R4 (1/2)**
>
> Dear R4,
>
> We sincerely appreciate your valuable comments and efforts helpful for improving our manuscript. We address each comment in detail, one by one as below.
>
> **(Q1) “The novelty of this paper is limited.”**
>
> (A1) We are aware that various replay schemes and ensembling schemes are widely used for RL. However, we believe that applying them in various fine-tuning setups is a novel contribution of our work. For instance, one of the first works in fine-tuning offline RL agents [1] only considers datasets of relatively small sizes (\~100K), in which case the disproportionate sizes of offline dataset and online samples is not a big issue. Accordingly, [1] uses a single replay buffer for fine-tuning. On the other hand, we also consider datasets of disproportionately large sizes (\~1M, \~2M), in which case it is necessary to sample a balanced mix of offline and online samples.
>
> Also, most distillation schemes focus on multi-task setups [2,3,4], and most ensemble schemes in RL focus on ‘train-from-scratch’ scenarios [5,6]. In light of this, we believe our ensemble distillation scheme is a novel contribution to the offline RL & fine-tuning literature.
>
> **(Q2) Other scheduling schemes for balanced replay.**
>
> (A2) We have additionally tried various other linear schedules and exponential schedules (see Appendix G, H, I of the revised manuscript). For one, we varied the initial fraction $\rho_{0}^{\texttt{on}}$ of online samples to take values from $\{0.25, 0.5, 0.75, 0.9\}$. Also, we performed search over the final timestep of the annealing schedule,  $t_{\texttt{final}}$. Finally, we also experimented with an exponential schedule. As shown in Figure 12, 13, 14, fine-tuning performance stays relatively robust to such different scheduling schemes in halfcheetah and most hopper datasets. However, performance can be sensitive for walker2d tasks, which is known to be a difficult control task.
>
> As pointed out by the R4, automatic adjustment of sampling ratio is an interesting and promising future direction. For example, one can adjust the sampling ratio based on how distribution of online samples is different from offline data distribution using density esimtation (see A3 and Figure 1b of the revised manuscript for more details). We would continue to add additional experiments under this scheme, if time permits.
>
> **(Q3) Formal description / analysis of distribution shift.**
>
> (A3) Note that offline dataset is generated according to the following distribution: $ p_{\texttt{data}}(s,a) = d^{\pi_{\beta}}(s)\pi_{\beta}(a|s), $
> where $\pi_{\beta}$ denotes the behavior policy, and $d^{\pi}$ the discounted marginal state distribution of $\pi$.
>
> Then distribution shift refers to the difference between $p_{\texttt{data}}(s,a)$ and $p_{\theta}(s,a)$, i.e., there exists a shift in both state distribution and state-conditioned action distribution.  We would like to point out that the distribution shift we tackle is different from the distribution shift usually referred to in the offline RL literature: the learned policy deviating from the behavior policy, i.e. $D(\pi_{\theta}(\cdot|s), \pi_{\beta}(\cdot|s))$ becoming large, for a given state $s\sim d^{\pi_{\beta}}$.
>
> As a measurement for the degree of distribution shift, we provide the histogram of log-likelihoods for offline and online samples (Figure 1b of the revised manuscript), obtained using VAE pre-trained using offline dataset  (also see our response A5 to R2 for more details). Also, we added the formal description of distribution shift in Section 3 of the revised manuscript.
>
> **(Q4) How does ensemble distillation tackle distribution shift?**
>
> (A4) Ensemble distillation reduces the policy variance, and thereby provides further robustness to Q-function approximation error. To show this, we performed an additional experiment: during training, given each observation in the sampled minibatch, we sampled 10 actions from the policy being trained, then measured the variance of Q values among these 10 actions. We averaged the variance in Q values across the minibatch samples. Figure 16 in Appendix K shows the result, where Q values for actions sampled by independent ensembles have much higher variance when compared to the ensemble distillation counterpart.

---

> > ### Author Response · Authors · 2020-11-19
> > **Response to R4 (2/2)**
> >
> > **(Q5) Details about t-SNE plot (Figure 1a)**
> >
> > (A5) Figure 1a visualizes the t-SNE embedding of state-action pairs (each datapoint representing the concatenation of a state and its corresponding action) observed online vs those contained in the offline dataset. In particular, online samples were collected by rolling out an offline RL (CQL) agent for several episodes until 1000 transitions were gathered. As for offline samples, trajectories were randomly drawn from the dataset until there were 1000 transitions sampled. This shows that the discounted stationary state-action distribution of $p_{\texttt{data}}(s,a)$ and that of $p_{\theta}(s,a)$ are distributed differently. We clarified this in Figure 1(a) and Section 3.
> >
> > **(Q6) Comparison of ‘ensemble-only’ with baselines**
> >
> > (A6) Following your suggestion, we provide additional experimental results that compares ‘ensemble-only’ to non-ensemble baselines (Appendix J). We observe that ‘ensemble-only’ does not significantly exhibit better performance than baselines, or even worsens the performance in a hard control task such as walker2d. However, as shown in Figure 3 from the original manuscript, we would like to emphasize that ensemble distillation significantly improves fine-tuning performance in most tasks when combined with balanced replay, by reducing the variance of policy.
> >
> > -----------------------------------------------------------------------------------------------------------------------------------------
> >
> > **REFERENCES**
> >
> > [1] Nair, Ashvin, et al. "Accelerating online reinforcement learning with offline datasets." arXiv preprint arXiv:2006.09359. 2020.
> >
> > [2] Rusu, Andrei A., et al. "Policy distillation." International Conference on Learning Representations. 2016.
> >
> > [3] Schmitt, Simon, et al. "Kickstarting deep reinforcement learning." arXiv preprint arXiv:1803.03835. 2018.
> >
> > [4] Traoré, René, et al. "DISCORL: Continual reinforcement learning via policy distillation." arXiv preprint arXiv:1907.05855. 2019.
> >
> > [5] Anschel, Oron, Nir Baram, and Nahum Shimkin. "Averaged-dqn: Variance reduction and stabilization for deep reinforcement learning." International Conference on Machine Learning. 2017.
> >
> > [6] Osband, Ian, et al. "Deep exploration via bootstrapped DQN." Advances in neural information processing systems. 2016.

---

### Official Review · AnonReviewer2 · 2020-10-29
**Inconclusive empirical evidence for BRED's performance and whether it directly addresses distribution shift**

**Rating:** 3
**Confidence:** 4

**Review:**

Summary
-------

Offline RL allows agents to be trained on static, offline datasets. If
the agent has already undergone offline training however, distribution
shift makes further online training difficult. The authors propose to
address this problem by keeping separate offline and online replay
buffers, and sampling different proportions of each. In addition, they
propose to ensemble a distillation policy. The proposed method is
evaluated on logged MuJoCo datasets answering whether BRED improves over
other fine-tuning methods and whether balancing or replay contribute
more to BRED's success.

Decision
--------

BRED is an interesting approach, but the experiments do not demonstrate
that it is better than the baseline. As such, my preliminary rating of
this paper is rejection. I appreciate that the sampling scheme proposed
is simple and well motivated to address the problem of distribution
shift. It is not clear whether the sampling scheme actually addresses
the problem of distribution shift however, as it gives the agent access
to additional experience. The distillation scheme also does not seem to
address distribution shift directly and the experiments do not
demonstrate that it helps performance outside of the Walker2d-random
task. Moreover, it is not clear whether the baselines are also
ensembles, which would make the experiment results less compelling.

Originality
-----------

The contributions are: a sampling scheme combining offline and online
samples, as well as a distillation scheme. The sampling scheme is
incremental and simple, but seems like a novel approach. Distillation on
the other hand, is well-explored in RL (Czarnecki et al. 2019) and does
not seem to contribute much to the problem addressed in the paper. Taken
together, the paper has limited novelty.

Quality and Clarity
-------------------

The paper is easy-to-follow and well-written. However, I have some
issues with the title. The title suggests that your aim is to address
distribution shift in online RL. However, distribution shift is a
problem due to offline training. Of course, the problem you are
addressing is further online training after offline training. Perhaps
the title should be reworded to make this more clear.

Strengths
---------

-   There is a clear exposition of the problem addressed, namely
    distribution shift. This is well motivated by the t-SNE
    visualization. Section 3 in particular highlights the correlation
    between distribution shift and the performance of offline RL
    algorithms.
-   BRED, the proposed method, is a straightforward solution via
    balancing offline and online replay buffers. In addition, the
    schedule for updating the proportion sampled of each replay buffer
    is well argued and also simple.
-   The results are clearly motivated by questions in the beginning of
    the section. Sensible baselines are used to compare the proposed
    method (BRED), and these are discussed and reasoned in detail.
    Furthermore, there is a good coverage of tasks (3 MuJoCo) and
    difficulties (random, medium, medium-replay, medium-expert).

Weaknesses
----------

-   While fine-tuning an offline RL agent can pose challenges, similar
    challenges can arise in supervised learning. For example, Ash and
    Adams (2019) note that in different regimes, warm-starting neural
    network training can hamper generalization. Overall, I feel that the
    correlation between distribution shift and online performance after
    offline training is not shown to be causal, but correlative.

-   It is unclear why distillation is included, as it does not directly
    address the problem of distribution shift in offline RL, outside of
    the intuition that it mitigates bad $Q$ estimates. This is not
    enough motivation however, and the experiment results (Figure 5
    and 6) does not show any significant benefit of ensemble
    distillation over independent ensembles (besides Walker2d-random).
    Furthermore, this may not be a fair comparison to other baseline
    methods which do not seem to be ensembles.

-   Despite a very well-motivated experiment section, the experimental
    conclusions are far too strong. The results do not back this up,
    when there is high overlap in the confidence intervals for many
    Figures (in each section). Furthermore, there are unexplained
    peculiarities in the results, such as very big drops in performance
    or the fact that the agent trained with the offline dataset is
    substantially better than the medium one.

Detailed Comments
-----------------

-   "pointed out that offline RL algorithms… are not amenable to
    fine-tuning, due to the difficulty of modeling the
    dataset-generating policy in the online setup."

    This is not clear to me, how does fine-tuning connect to modelling
    the dataset-generating policy? Furthermore, how does the
    dataset-generating policy (I assume this is the behavior policy)
    connect to online RL?

-   Figure 1b: the agent is trained on the offline dataset and then that
    dataset is thrown away for the online agent, whereas the agent in
    red gets to keep using it. This does show the difficulty of
    fine-tuning but does not provide conclusive evidence that
    distributional shift is the cause. Fine-tuning can be difficult even
    in supervised learning (Ash and Adams 2019).

-   "On the other hand, when using online samples exclusively, the agent
    is exposed to unseen samples only, for which Q function does not
    provide a reliable value estimate. This may lead to bootstrapping
    error, and hence a dip in performance as seen in Figure 1b."

    Can't this be shown experimentally in an environment to provide
    conclusive evidence for distribution shift being the cause of the
    dip in performance?

-   Figure 3: It seems that halfcheetah-random results in a better
    policy than halfcheetah-medium. This does not make sense to me. In
    addition, there is a strange dip in performance for CQL-ft in Figure
    3g. Many of the figures have overlapping confidence intervals
    (although the shaded region is only the standard deviation, the
    corresponding confidence interval would be approximately the
    standard deviation. Anyway these should be confidence intervals
    rather than standard deviation). While BRED does perform
    statistically significantly better in some tasks, I don't think the
    comparison is fair due to ensembling used only for BRED.

-   Figure 4: Going from figure 4a to 4b, balanced replay is shown to be
    more beneficial for a dataset generated by a random behavior policy.
    This seems to suggest that the main benefit of BRED is artificial
    exploration. Counter-intuitvely, this trend is reversed in Figure
    4c. Moreover, the online-only curve is very different in Figure 4c
    than 4a and 4b, and to my understanding there should not be any
    difference. Due to the noisiness of the curves, its hard to make any
    straightforward conclusion from these results.

-   Figure 5 and 6: Again, there is no statistical difference in medium
    and medium-replay between independent ensembles and a distillation
    ensemble.

Minor Comments
--------------

-   Equation 3: missing brackets on the left side.


Ash, Jordan T., and Ryan P. Adams. 2019. “On Warm-Starting Neural Network Training.” *arXiv:1910.08475*.
<http://arxiv.org/abs/1910.08475v2>.

Czarnecki, Wojciech Marian, Razvan Pascanu, Simon Osindero, Siddhant M. Jayakumar, Grzegorz Swirszcz, and Max Jaderberg. 2019. “Distilling Policy Distillation.” *arXiv:1902.02186*.
<http://arxiv.org/abs/1902.02186v1>.

---

> ### Author Response · Authors · 2020-11-19
> **Response to R2 (1/3)**
>
> Dear R2,
>
> We sincerely appreciate your valuable and insightful comments. We found them extremely helpful for improving our manuscript. We address each comment in detail, one by one below.
>
> **(Q1) Clarification on the meaning of title.**
>
> (A1) We agree with the author that the title may be misleading, for we mainly deal with fine-tuning offline RL agents, not randomly initialized agents. Accordingly, we changed the title from ‘Addressing Distribution Shift in Online Reinforcement Learning with Offline Datasets’ to ‘Addressing Distribution Shift in Offline-to-Online Reinforcement Learning’.
>
> **(Q2) Clarification on dataset-generating policy and its relationship to online RL.**
>
> (A2) Thank you for pointing out a possibly confusing point. A prior work [1] pointed out that it is not feasible to extend such offline RL methods as BCQ [2] and BEAR [3] to the fine-tuning setup. This is because these early methods require modeling the policy used to generate the offline dataset, i.e. behavior policy/dataset-generating policy, and training such density models in the online fine-tuning setup is challenging [1]. Instead, we consider building upon a more recent offline RL algorithm, conservative Q-learning (CQL) [4], which does not require modeling the behavior policy. Accordingly, our method does not model the behavior policy. We clarified this in Section 1.
>
> **(Q3) Is instability in fine-tuning due to distribution shift?**
>
> (A3) We would like to point out that there have been several prior works where fine-tuning has been successfully applied for robot learning [5,6]. These works rely on an ample amount of diverse offline data, whereas we considered various scenarios where such large and diverse dataset may not be available, in which case distribution shift may hamper the fine-tuning process.
>
> Also, we would like to emphasize that fine-tuning a Q-learning RL agent poses a unique challenge when compared to supervised learning, for it involves bootstrapping - a process especially vulnerable to distribution shift. Indeed, a prior work [1] trained a SAC agent from scratch, but with a pre-populated replay buffer, i.e., an agent that suffers from distribution shift, but not from the difficulty of fine-tuning a neural network. Thusly trained SAC agent is shown to perform worse than a SAC agent trained from scratch with an empty replay buffer. This observation shows that distribution shift is indeed a non-trivial challenge when training an RL agent.
>
>
> **(Q4) Justification of ensemble distillation.**
>
> (A4) Ensemble distillation provides further robustness to Q-function approximation error, by reducing policy variance. In order to demonstrate this, we conducted an additional experiment: for each update during training, given an observation in the sampled mini-batch, we sampled 10 actions from the policy being trained, then measured the variance of Q values among these 10 actions. Then we averaged the variance in Q values across the sampled minibatch, i.e., $\mathbb{E}_{s\sim \mathcal{B}, a\sim \pi(\cdot|s)}\big[ Q_\theta(s,a) \big]$. As shown in Appendix K, Figure 16, Q values for actions thusly sampled are of much higher variance for the independent ensemble learner. This shows that by training a single distilled policy, we obtain a stability gain.
>
>
>
> **(Q5) Direct evidence of distribution shift.**
>
> (A5) In order to demonstrate the distribution shift in a more direct manner, we additionally trained a variational autoencoder (VAE) and measured the log-likelihoods of offline and online samples (Figure 1b of the revised manuscript). We observe that offline and online samples follow different distributions, which evidences that there exists a distribution shift (see our response A3 to R4 for more formal description of distribution shift). Specifically, we first split a given offline dataset into a training dataset and validation dataset. Then, we trained a VAE on the training offline dataset, and measured log-likelihood for (1) offline samples from the held out validation dataset and (2) online samples gathered by the offline agent. We included the experimental detail in Section 3 of the revised manuscript also.
>
> In order to show the harmful effects of distribution shift more directly, we conducted additional experiments with walker2d tasks. Similar to [7], we compared the average Q value estimate over 1000 state-action pairs sampled from the replay buffer, and the average of their true Q values. As shown in Figure 17 from Appendix L of the revised manuscript, we see that in three out of four tasks (walker2d-random, walker2d-medium-replay, and walker2d-medium-expert), Online-only agents suffer from severe overestimation bias and high variance, especially during the early fine-tuning stage. Experimental details are also provided in Appendix L of the revised manuscript.

---

> > ### Author Response · Authors · 2020-11-19
> > **Response to R2 (2/3)**
> >
> > **(Q6) Comparative evaluation with ensemble baselines.**
> >
> > (A6) As R3 and R4 pointed out, we believe our experiments effectively demonstrate the strength of our method. To address the concern with non-ensemble baselines, we performed additional experiments and updated all baselines with ensemble baselines, i.e., ensemble of AWAC, BCQ-ft, CQL-ft, and SAC, respectively (Figure 3 of the revised manuscript). We observe that BRED still outperforms baseline methods in most tasks in terms of both training stability and sample-efficiency.
> >
> > **(Q7) Better asymptotic fine-tuning performance on halfcheetah-random.**
> >
> > (A7) One possible explanation for this is that random exploratory data provides a wide support of state-action space coverage, which is hypothesized to result in better Q-learning [8]. This effect does not appear as pronounced in other environments (hopper, walker2d), where the fine-tuned agent reaches an expert-level performance quite early in training.
> >
> > **(Q8) Performance drop on halfcheetah-medium-replay (Figure 3g)**
> >
> > (A8) As for halfcheetah-medium-replay from Figure 3g, there was a temporary performance drop in one out of four seeds, possibly due to the policy finding a bad local optimum.
> >
> > **(Q9) Explanation of trends for different datasets in the ablation study on balanced replay (Figure 4)**
> >
> > (A9) Figure 4 shows the benefit of BRED over (1) online-only, and (2) uniform replay. As we explained in Section 3, online-only uses online samples gathered exclusively for fine-tuning an offline RL agent, hence it suffers from severe bootstrap error. Uniform replay, on the other hand, draws samples uniformly at random, and hence cannot update the agent at novel states and actions encountered online in a timely manner.
> >
> > Indeed, for the walker2d-random task, online-only agent fails to improve due to severe bootstrap error at states and actions encountered online, whereas uniform replay agent hardly improves for it does not see enough near-on-policy samples. For similar reasons, in walker2d-medium, online-only agent suffers from unstable training, while uniform replay agent regresses to the average performance observed in the offline dataset. For walker2d-medium-replay, uniform replay improves more steadily, for the size of dataset is about 10x smaller than that of walker2d-random and walker2d-medium, but the improvement is still slow.
> >
> > Also, in Figure 4, since online-only agents are updated using only online samples after they have already been trained on the offline datasets using CQL, performances of online-only agents in random, medium, medium-replay are different, as base offline agents are trained with different datasets. We clarified the meaning of ‘online-only’ in Figure 4 and Section 6.3 in the revised manuscript.
> >
> > **(Q10) Different performance of ‘online-only’ agents on different datasets.**
> >
> > (A10) We first clarify that, as explained in Figure, online-only agents are first trained with offline datasets of different qualities (random, medium, medium-replay), respectively, then are fine-tuned only using samples collected online. Hence, performances of online-only agents are different as the initial offline policies are different.
> >
> > For instance, online-only agents trained with medium and medium-replay datasets, respectively, achieve better initial & fine-tuning performance than the online-only agent trained with random dataset, for they start with more performant offline policies. Online-only agent trained with medium dataset performs better than the agent trained with medium-replay dataset, possibly because medium dataset contains 10x more transitions than the medium-replay dataset. We clarified this in Section 6.3 and Figure 4 of our revised manuscript.

---

> > > ### Author Response · Authors · 2020-11-19
> > > **Response to R2 (3/3)**
> > >
> > > **(Q11) Messages from ablations studies on ensemble size / distillation (Figure 5,6)**
> > >
> > > (A11) As shown in Figure 5a, 5b, independent ensembles suffer from high variance in performance, indicating unstable learning. As mentioned in (A3), our additional experiment (Appendix K) is further evidence that distillation reduces policy variance, and hence benefits from a more accurate policy update.
> > >
> > > Also, Figure 6 is an ablation on ensemble size, and the results indicate that ensemble indeed provides performance gain, as the ensemble size increases. No comparison of BRED to independent ensembles is being made in this figure.
> > >
> > > **(Q12) Confidence interval instead of reporting standard deviations.**
> > >
> > > (A12) Thank you for the suggestion. We updated all figures in the revised manuscript to report 95% confidence intervals instead of standard deviations.
> > >
> > > **Minor comment:**
> > >
> > > We thank the reviewer for pointing out a typo. We corrected it.
> > >
> > > -----------------------------------------------------------------------------------------------------------------------------------------
> > >
> > >
> > > **REFERENCES**
> > >
> > > [1] Nair, Ashvin, et al. "Accelerating online reinforcement learning with offline datasets." arXiv preprint arXiv:2006.09359 (2020).
> > >
> > > [2] Fujimoto, Scott, David Meger, and Doina Precup. "Off-policy deep reinforcement learning without exploration." International Conference on Machine Learning. 2019.
> > >
> > > [3] Kumar, Aviral, et al. "Stabilizing off-policy q-learning via bootstrapping error reduction." Advances in Neural Information Processing Systems. 2019.
> > >
> > > [4] Kumar, Aviral, et al. "Conservative Q-Learning for Offline Reinforcement Learning." Advances in Neural Information Processing Systems. 2020.
> > >
> > > [5] Kalashnikov, Dmitry, et al. "Qt-opt: Scalable deep reinforcement learning for vision-based robotic manipulation." Conference on Robot Learning. 2018.
> > >
> > > [6] “Never Stop Learning: The Effectiveness of Fine-Tuning in Robotic Reinforcement Learning.” Conference on Robot Learning. 2020.
> > >
> > > [7] Fujimoto, Scott, Herke Hoof, and David Meger. "Addressing Function Approximation Error in Actor-Critic Methods." International Conference on Machine Learning. 2018.
> > >
> > > [8] Fu, Justin, et al. "Diagnosing bottlenecks in deep q-learning algorithms." International Conference on Machine Learning. 2019.

---

> > > > ### Comment · AnonReviewer2 · 2020-11-23
> > > > **Thank you for the very detailed reply and the updated draft**
> > > >
> > > > Thank you for the very detailed reply and the updated draft. While the
> > > > changes have strengthened the paper, I still feel that there is a
> > > > disconnect between the goals, contributions and findings. The goal main
> > > > goal is to obtain good fine-tuning performance and sample efficiency.
> > > > The contribution to achieve this goal is BRED (which combines a sampling
> > > > scheme with ensemble distillation). Yet the findings do not demonstrate
> > > > that the contributions achieve this goal.
> > > >
> > > > Your changes have made the challenge of distribution shift and the
> > > > motivation much more clear. I especially like that you use a VAE to
> > > > demonstrate distribution shift in Figure 1. Echoing R1, online
> > > > fine-tuning is difficult to motivate. Either you have a simulator to
> > > > begin with and can train online from the beginning or you do not have a
> > > > simulator and are unable to fine-tune. It is not immediately obvious
> > > > that fine-tuning (i.e. online training after offline training) is better
> > > > than interleaving online and offline training. The authors point to
> > > > literature in the robotics literature that helps motivate this. I remain
> > > > skeptical that this problem is more important, or distinct from,
> > > > sim2real. However, I do understand the reasons for approaching the
> > > > problem this way.
> > > >
> > > > One of the main contributions, ensemble distillation, is not well
> > > > motivated to address the problem of online fine-tuning. There is no
> > > > doubt that it helps performance in Walker2d-random. However, the reason
> > > > for that is the reduced variance of the action-value estimate. This
> > > > explanation is orthogonal to the problem being addressed, and would be
> > > > equally helpful outside of this context. Lastly, you state that the
> > > > baselines are ensembled while BRED uses ensemble distillation. You also
> > > > state that ensemble distillation is more effective than independent
> > > > ensembles. I would then think that the most fair comparison would also
> > > > use ensemble distillation, not independent distillation, for the
> > > > baselines.
> > > >
> > > > This leads into the empirical assessment. Outside of Walker2d-random,
> > > > Figures 4, 5 and 6 do not convince me of the general effectiveness of
> > > > BRED. In Figure 4 for example, uniform sampling amounts to mostly
> > > > offline sampling (because the dataset has 10<sup>6</sup> samples) while
> > > > online sampling throws away data (the offline dataset). So, it makes
> > > > sense that weighing the samples from both sources will perform better.
> > > > This is only clearly demonstrated in Walker2d-random, as the confidence
> > > > intervals look to overlap in medium and medium-replay. The same is true
> > > > for Figure 5, which shows that ensemble distillation is not
> > > > statistically different from independent distillation in medium and
> > > > medium-replay. Figure 6 also does not demonstrate statistical difference
> > > > between ensemble sizes for walker2d-medium and walker2d-medium replay.
> > > > In the presence of these findings, it does not seem that the main
> > > > contributions of BRED (sample weighting and ensemble distillation) are
> > > > as performant as claimed. Even in Figure 3, the only statistically
> > > > significant conclusion to be made for BRED is that it outperforms the
> > > > baselines on half-cheetah and walker2d-random. While you have stated
> > > > that these baselines use ensembles, I am unsure if they use distillation
> > > > ensembles or independent ensembles.
> > > >
> > > > Since the empirical assessment and ensemble distillation are core parts
> > > > of this paper, I do not think it is possible to address these concerns
> > > > in a revision.

---

> > > > > ### Author Response · Authors · 2020-11-24
> > > > > **Response to R2**
> > > > >
> > > > > Many thanks for your response before the discussion phase ends.
> > > > >
> > > > > **(Q1) Motivation for online fine-tuning**
> > > > >
> > > > > (A1) Although all our experiments are based on simulated MuJoCo environments, we believe that our method would also be applicable to real-world robotics, given an appropriate offline dataset and a real-world applicable offline RL algorithm. In such a real-world scenario, it is sensible to assume that a dataset consisting of optimal transitions is not always available, in which case the proposed method would make sample-efficient fine-tuning possible. We agree that sim2real is an important problem, but our focus is not on transferring the policy learned in simulation to that in the real-world, and is orthogonal to the problem setup of sim2real.
> > > > > Also, we would like to clarify that offline training then fine-tuning corresponds to the initial iteration of interleaving offline and online training. In our work, we observe that online training followed by offline training is challenging due to distribution shift. We believe addressing this issue will help scale to a more general scenario of interleaving offline and online RL training.
> > > > >
> > > > > **(Q2) Does ensemble distillation directly tackle the fine-tuning setup?**
> > > > >
> > > > > (A2)  Ensemble distillation is more helpful in the fine-tuning setup than in the train-from-scratch setup. This is because in the train-from-scratch setup, variance in action-value estimates with respect to sampled actions may actually benefit exploration. In the fine-tuning setup, however, our primary goal is to stabilize policy learning, in which case reduced variance due to ensemble distillation is beneficial.
> > > > >
> > > > > **(Q3) Which ensemble method was used for baseline methods?**
> > > > >
> > > > > (A3) We clarify that independent ensembles are used for the baselines. We would like to emphasize that ensemble distillation is our proposed component of BRED, and hence was not applied to the baseline methods. Nonetheless, one can expect that the baseline methods, i.e., BCQ-ft and AWAC, would not benefit much from ensemble distillation. As we mentioned in the previous response, AWAC relies on regression for updating the policy, rather than optimizing the policy parameters in the direction of increasing Q values. In other words, AWAC does not sample actions from the policy during policy update, hence does not suffer from policy variance. On the other hand, the policy from BCQ-ft has the minimal role of perturbing action samples from the behavior model (modeled as a VAE), and it is unlikely that ensemble distillation would lead to any performance gain. Overall, the trends would remain unchanged. However, please understand that we cannot provide additional experimental results due to limited time.
> > > > >
> > > > > **(Q4) Experiment results are not convincing**
> > > > >
> > > > > (A4) As for Figure 3, we agree with the reviewer that BRED does not outperform all baselines in every single task considered. However, as mentioned in Section 6.2, we would like to point out that BRED is the only method that consistently performs well across all tasks considered. In contrast, AWAC shows (1) suboptimal asymptotic performance (all halfcheetah tasks, hopper-random, all walker2d tasks except for medium-expert), or (2) training instability (hopper-medium-expert, walker2d-medium-expert). Similarly, CQL-ft shows (1) suboptimal asymptotic performance (all halfcheetah tasks except medium-expert, walker2d random and medium), or (2) training instability (all hopper tasks, walker2d-medium). BCQ-ft shows suboptimal asymptotic performance in all but the hopper-medium-expert task. In light of this, we believe Figure 3 clearly demonstrates the effectiveness of BRED.

---

### Author Response · Authors · 2020-11-19
**Summary of revisions**

Dear reviewers,

We sincerely appreciate your insightful comments and constructive suggestions to help us improve the manuscript. We are grateful for all positive comments: well-written (R1, R2, R3, and R4), well-motivated (R2, R4), and good experimental results (R3, R4).

In response to the questions and concerns you raised, we have carefully revised and improved the manuscript with the following additional experiments and discussions:

- Formal definition of distribution shift (Section 3)
- Empirical evidence of distribution shift (Figure 1b)
- Performance comparisons with ensemble baselines and offline RL agents (Figure 3)
- Empirical justification of balanced replay for addressing distribution shift (Appendix L)
- Empirical justification of ensemble distillation for addressing distribution shift (Appendix K)
- Analysis on the effects of different online RL algorithms on the overall fine-tuning performance (Appendix C)
- Analysis on sensitivity of the proposed balanced replay scheme to hyperparameter selection (Appendix G, H)
- Additional ablation study of the proposed balanced replay with exponential annealing schedule (Appendix I)
- Clarification on prior works (Section 1, 5)

The revisions made are marked with “red” in the revised manuscript.

We also appreciate your continued effort to provide further feedback until the very end of response/discussion phase. We will make sure to reflect the comments in the final version.

Best regards,

Authors.

---

### Decision · Program_Chairs · 2021-01-07
**Final Decision**

**Decision:**

Reject

**Comment:**

This paper addresses an distribution shift and biased Q-values that happens when offline agents are finetuned in an online manner. The final revision of the paper is very well written and easy to understand. The proposed method in the paper is interesting, and aiming to address an important issue in RL. The proposed method involves a combination of two well-known methods in RL to tackle the distribution shift issue, the paper first suggests to use a balanced replay mechanism a replay for online experiences and another one for the offline. The second improvement is coming from the ensemble distillation.

It seems like in the light of the reviews, the authors have improved manuscript. However, I would like to recommend the paper for rejection. I would like the authors to do further experiments on the individual components of the algorithms, for example what if we run all the experiments only with BR or only using ED how would the performance change. How much improvement is coming from each one of those individual components? As it stands, it is not clear to me right now, and the proposed solution looks a bit complicated and hacky.

The balanced replay mechanism is very similar to the replay approaches that are used for learning from demos methods like R2D3 [1] and DQfD. Also the ensemble distillation approach is very akin to RAND [2] and distillation approaches that are used in lifelong learning algorithms. It is not clear, why it is that important for offline RL. It should potentially improve online RL as well, perhaps some experiments on online RL would be interesting.

Nevertheless, I think the paper is very interesting and attempting to address a very important problem in RL. I would recommend the authors to resubmit the paper to a different venue after doing some small changes on it.

[1] Gulcehre, C., Le Paine, T., Shahriari, B., Denil, M., Hoffman, M., Soyer, H., ... & Barth-Maron, G. (2019, September). Making Efficient Use of Demonstrations to Solve Hard Exploration Problems. In International Conference on Learning Representations.

[2] Burda, Y., Edwards, H., Storkey, A., & Klimov, O. (2018). Exploration by random network distillation. arXiv preprint arXiv:1810.12894.